

# Covariant entanglement wedge cross-section, balanced partial entanglement and gravitational anomalies

Qiang Wen[1,2★] and Haocheng Zhong[2†]

**1** Shing-Tung Yau Center of Southeast University, Nanjing 210096, China
**2** School of Mathematics, Southeast University, Nanjing 211189, China

★ wenqiang@seu.edu.cn, † zhonghaocheng@outlook.com

## Abstract

The balanced partial entanglement (BPE) was observed to give the reflected entropy and the entanglement wedge cross-section (EWCS) for various mixed states in different theories [1, 2]. It can be calculated in different purifications, and is conjectured to be independent from purifications. In this paper we calculate the BPE and the EWCS in generic covariant scenarios in two-dimensional CFTs with and without gravitational anomalies, and find that they coincide with the reflected entropy. In covariant configurations we determine the partition for the purifying system with the help of the gravitational anomalies, and we extend our discussion to topological massive gravity (TMG). We give the first prescription to evaluate the entropy quantity associated to the EWCS beyond Einstein gravity, i.e. the correction to the EWCS from the Chern-Simons term in TMG. Apart from the gravity theory and geometry, further input from the mixed state should be taken into account.



# 1 Introduction

In recent years, concepts and ideas of quantum information sciences have been applied to holographic theories between gravity and quantum field theories, which strongly suggests that gravity actually emerges from the entanglement structure of the dual quantum field theory. On the other hand, investigations for quantum gravity also stimulate new understanding in quantum information sciences. One example is the measure of correlations in mixed states. This topic has received lots of inspiration from high energy physics and has attracted considerable attention recently.

The entanglement entropy is one of the most important and common measure for quantum entanglement. For a bipartite system $AB \equiv A \cup B$ in a pure state $|\Psi\rangle$, the entanglement entropy $S_A$ of the subsystem $A$ is defined as the von Neumann entropy of the reduced density matrix $\rho_A = \mathrm{Tr}_B |\Psi\rangle\langle\Psi|$ of $A$. The entanglement entropy $S_A$ captures the amount of entanglement between $A$ and $B$. Nevertheless, if we want to measure the correlation between $A$ and $B$ when $AB$ is in a mixed state, the entanglement entropy is no longer a good measure. This can be clearly shown by introducing an auxiliary system $O$ such that $ABO$ makes up a pure state, which is called a purification of $AB$. In such a purification $S_A$ captures the amount of entanglement between $A$ and $BO$, rather than the one between $A$ and $B$.

Indeed, it is hard to define a simple quantity that may capture the "amount of entanglement" in mixed states, and even harder to clarify what does it mean by the "amount of entanglement" in mixed states. Nevertheless, several different measures have been proposed, which collect different types of correlations and are used for different purposes. Many of the measures are defined in terms of an optimization problem, which makes them formidable to calculate except in some extremely simple systems. These measures include the entanglement of formation, the entanglement of distillation, the entanglement of purification (EoP) and so on. For example, consider an arbitrary auxiliary system $A'B'$ which purifies $AB$, the EoP is defined to be the minimal value of $S_{AA'}$ among all the possible purifications and all the possible partition $A'B' = A' \cup B'$ for the purifying system [3].

There are also measures defined directly on the density matrix $\rho_{AB}$ of the mixed state and being calculable. One important example is the logarithmic entanglement negativity [4–8], which is defined as $\mathcal{E}(A,B) = \ln \mathrm{Tr}|\tilde\rho_{AB}|$, where $\tilde\rho_{AB}$ is the partial transposed $\rho_{AB}$ on the $A$ indices. Another interesting measure is the reflected entropy [9], which is defined to be $S_R(A,B) = S_{AA'}/2$, where $S_{AA'}$ is evaluated in the canonical purification of the mixed state. Note

that, throughout this paper the reflected entropy we defined is *half* of its original definition [9]. Since the negativity and the reflected entropy are both calculable, they are wildly explored in many condensed matter systems, as well as holographic theories. Nevertheless, it is still hard to perform their calculation in quantum field theories.

In AdS/CFT, the reduced density matrix of a boundary region *AB* duals to a specific bulk region called the entanglement wedge $\mathcal{W}_{AB}$, which is a co-dimension-one surface bounded by *AB* and corresponding Ryu-Takayanagi (RT) surface $\mathcal{E}_{AB}$[1],

$$\partial \mathcal{W}_{AB} = AB \cup \mathcal{E}_{AB}. \tag{1}$$

Inside the entanglement wedge, the minimal cross-section (EWCS) $\Sigma_{AB}$ that separates *A* from *B* is a special geometric quantity that could be considered as a natural measure of the entanglement between *A* and *B*, hence it should be dual to a measure of mixed state correlations. It is an important topic to explore as the duality will extend the correspondence between the geodesics anchored on the boundary and the entanglement entropy, to the correspondence between geodesic chords and mixed state correlations. It can be considered as a finer version of the RT formula.

Interestingly, multiple of the above-mentioned measures for mixed state correlations are claimed to be dual to the area of the EWCS, which is similar to the way the RT surface captures the entanglement entropy. These measures include the EoP [3,10,11], the entanglement negativity [12–14], the reflected entropy [9], the "odd entropy" [15], the "differential purification" [16], the entanglement distillation [17,18]. Recently, the duality for the EWCS goes beyond AdS/CFT. Based on the geometric picture [19–22] for the holographic entanglement entropy in several holographic models beyond AdS/CFT, the EWCS is generalized to the cases of 3-dimensional flat holography [23] and (warped) $AdS_3$/warped CFT correspondence [24]. Furthermore, the correspondence between the negativity, the reflected entropy and the EWCS is confirmed by explicit calculations on both sides. See [25–27] for more relevant discussions.

All the above measures are defined in different ways hence should capture different types of correlations. For some of the measures, it is hard to prove or disprove their duality with the EWCS due to the difficulty to conduct explicit calculations. For those calculable measures, explicit calculations can only be conducted and found matched with the EWCS in several special cases. The evidence for generic scenarios, and the logic indicating that these measures of correlations lead to a clear geometric picture is still missing. It is also possible that in holographic theories several of them coincide with each other and can be given by the EWCS. So far there are more evidences for the duality between the reflected entropy and the EWCS, but it is still hard to confirm or exclude other proposals.

In this paper we focus on a new quantity named the *balanced partial entanglement* (BPE) [1,2], which is the *partial entanglement entropy* (PEE) [28,29] that satisfies certain balance conditions. The PEE satisfies the key property of additivity, as well as all the properties (inequalities) satisfied by the mutual information. It is proposed in [28,29] to give a fine description for the spatial structure of quantum entanglement for quantum systems. Furthermore, its correspondence to the geodesic chords in the context of holography is also proposed and confirmed in [28,29] using the geometric picture in the gravity side. For these reasons, it is possible to extract correlations in mixed states that correspond to the EWCS from the PEE. This is explicitly done in [1,2], where the BPE is defined in terms of the PEE with no reference to the geometric picture in the bulk. We will give an explicit introduction for the PEE and the BPE in the next section. Before that, let us summarize the advantages and dis-advantages of the BPE compared with the above-mentioned measures.

---

[1]Originally the entanglement wedge is the causal development of this co-dimension one surface.

**Advantages of the BPE**

- The BPE can be defined in any purification of the mixed state $\rho_{AB}$.

- The BPE can be easily calculated. More explicitly it is just a linear combination of certain subset entanglement entropies in $ABA'B'$, where the partition of the purifying system $A'B'$ is determined by the balance conditions, i.e. solving several linear equations.

- The BPE is a quantum information quantity which can be applied to general quantum systems. In holographic theories, the BPE exactly matches with the EWCS at order $c$.

- In the canonical purification, it is easy to see that the BPE coincides with the reflected entropy, hence could be considered as a generalization for the reflected entropy.

- Most of the measures are mainly explored in the static configurations while the BPE can be naturally extended to covariant scenarios, which is one of the main topics of this paper.

**Disadvantages of the BPE**

- Since the BPE can be defined in general purifications, we need to demonstrate that the BPE is purification independent if we claim it to be an intrinsic quantity. Although the independence from purifications for the BPE has passed several non-trivial tests in [2], it has not been proved in generic configurations.

- So far, calculations of the BPE are confined in two-dimensional theories. The reason is that our evaluation for the PEE is mostly well-studied in two-dimensional theories via the so-called *ALC* proposal (see the paragraphs near (8) for a brief introduction), which cannot be generalized to higher dimensions straightforwardly.

On the gravity side, the EWCS is mostly explored in AdS$_3$/CFT$_2$ for static configurations. Its extension to covariant configurations (see for example [27,30]) and in higher dimensions [2] remains rarely explored. Another important direction to generalize our understanding of the EWCS is to consider gravities beyond Einstein gravity, for example the gravities with high derivative corrections, and topological massive gravity (TMG) with an additional Chern-Simons (CS) term described by (56). So far, such generalizations have been extensively studied for the RT surfaces, see [38–42] for the higher derivative gravities and [43] for TMG. Similar generalization for the EWCS should also be very interesting and important. In this paper we will give explicit discussions on the EWCS in TMG.

**Main tasks**

In this paper our first task is to investigate the BPE and the EWCS in generic covariant scenarios in two-dimensional CFTs. This is meaningful because the measures proposed to capture mixed state correlations are rarely explored in covariant configurations. We find that the BPE, the EWCS and the reflected entropy exactly match with each other. In covariant configurations the partition points of $A'B'$ can vary in the two-dimensional spacetime rather than settled on a time slice, we therefore need two parameters to fix one partition point. Then we come up with the problem that the number of the balance conditions is less than the number of the parameters we need to fixed all the partition points. We will solve this problem by considering the CFT$_2$ with gravitational anomalies where the left and the right moving central charges are different. In these theories the balance conditions should hold for both the right moving

---

[2]See [31–37] for calculations in higher dimensional Einstein gravity with matter, excitation or time evolution.

and the left moving sectors. This will double the number of constraints from the balance conditions, which is enough to fix all the partition points in $A'B'$. Another way to fix the partition points is to extend the EWCS in the bulk into a geodesic anchored on the boundary, then the points where the extended geodesic reaches the boundary are the partition points for $A'B'$. As expected we find that the BPE exactly matches with the reflected entropy calculated in [44], as well as the EWCS in covariant configurations.

Our second task is to evaluate the correction to the EWCS from the CS term in TMG. For Einstein gravity, entropy quantities are proportional to the area of extremal surfaces while for generic gravities with covariant Lagrangian constructed from metric, the entropy receives corrections to the area term which can be calculated by the Wald formula [45–47]. The corrections to the black hole entropy and the entanglement entropies have been extensively studied in literature (see [43,48] in the case of TMG). Nevertheless, for the EWCS which is assumed to capture certain mixed state correlations, similar to the way that the RT formula or Bekenstein-Hawking formula does for pure states, the correction from the additional terms beyond the Einstein-Hilbert action in the action has not been explicitly explored before[3]. In TMG, the EWCS without the correction from the CS term will no longer match with its field theory counterpart, i.e. the reflected entropy and the BPE. In this paper, we will give a simple and natural prescription to embed the correction to the EWCS geometrically.

## 2 Backgrounds

### 2.1 A brief introduction to the balanced partial entanglement

The definition of the BPE is based on the concept of the PEE [28, 29, 49], which is a quasi-local measure of entanglement between two space-likely separated regions. For example if we denote the two regions as $A$ and $B$, the PEE between them is denoted as $\mathcal{I}(A, B)$[4]. The *entanglement contour* [50], which is the derivative version of the PEE, has been explored in the Gaussian state of several free lattice models [50–57] and is used to characterize the density distribution of the entanglement entropy $S_A$ for the region $A$. It has been studied to capture evolution of the entanglement distribution [50, 56, 58, 59], in holographic theories [21, 28, 60–62] and quantum information theories [60, 61, 63]. In the context of holography, the entanglement contour has a geometric interpretation, which is a fine correspondence between the points on the boundary interval $A$ and the points on the RT surface $\mathcal{E}_A$ [21, 28]. More interestingly, such a fine correspondence indicates that a segment on the RT surface will corresponds to the PEE between $A_i$, a subset of $A$, and $\bar{A}$. This observation inspires our later study on the EWCS in terms of the PEE [1, 2], since the EWCS is also a geodesic chord in the bulk.

The PEE satisfies all the properties of the mutual information. For example, it is also symmetric under permutation and reproduces the entanglement entropy $S_A$ when $AB$ is in a pure state (normalization),

$$\mathcal{I}(A, B) = \mathcal{I}(B, A), \qquad S_A = \mathcal{I}(A, B)|_{B \to \bar{A}}. \tag{2}$$

Unlike any other measure, the PEE is featured by the key property of additivity, i.e.

$$\mathcal{I}(A, BC) = \mathcal{I}(A, B) + \mathcal{I}(A, C). \tag{3}$$

Due to the permutation symmetry and the additivity property, one can further split the regions $A$ and $B$ into smaller and smaller pieces until single sites, hence the PEE is just given by a

---

[3]See [44] for an earlier attempt without justification.
[4]Note that the symbol we used for PEE looks quite similar to the mutual information $I(A,B)$.

collection of PEE between two sites,

$$\mathcal{I}(A,B) = \sum_{i \in A, j \in B} \mathcal{I}(i,j), \tag{4}$$

where $\mathcal{I}(i,j)$ means the PEE between the $i$th site and $j$th site. Furthermore, according to the normalization, the entanglement entropy can be evaluated from the PEEs

$$S_A = \mathcal{I}(A,\bar{A}) = \sum_{i \in A, j \in \bar{A}} \mathcal{I}(i,j), \tag{5}$$

as well as the *entanglement contour*

$$s_A(i) = \mathcal{I}(i,\bar{A})\Big|_{i \in A}, \tag{6}$$

which is a function defined on $A$ that characterizes how much entanglement each site inside $A$ contributes to the entanglement entropy $S_A$. In continuous systems like quantum field theories, the summation becomes integration. When we consider the contribution from a subregion $A_i$ to $S_A$, we just collect all the contribution from the sites inside $A_i$. Usually we write this contribution as $s_A(A_i)$, it is indeed a PEE between $A_i$ and $\bar{A}$,

$$s_A(A_i) \equiv \mathcal{I}(A_i,\bar{A}) \equiv \mathcal{I}_{A_i \bar{A}}. \tag{7}$$

Hereafter we interchangeably use the notations between $s_A(A_i)$ and $\mathcal{I}(A_i,\bar{A})$.

One may ask whether all the entanglement quantities can be evaluated based on the PEE. The answer is definitely no. In [49, 50], a set of physical requirements that the PEE should respect is classified, which include the aforementioned properties like normalization, permutation symmetry, additivity and other four requirements[5]. One may worry that the solutions to the set of requirements are not unique hence the concept of the PEE or entanglement contour is not well-defined. It is true that the uniqueness of the PEE has not been confirmed in generic theories. However, it can be confirmed at least in the following two configurations:

1. Generic two-dimensional theories where all the degrees of freedom lives with a natural order along the spatial direction [28, 29, 58] and highly symmetric higher dimensional theories where the contour function only depends on one coordinate which we call the quasi-one-dimensional configurations [60];

2. Higher dimensional theories with Poincaré symmetry [49].

We stress that the evaluation of the entanglement entropies following (5) only works for connected regions even for the above two types of configurations, for disconnected regions it is not solved in the context of the PEE.

Although, the mathematical definition for the PEE based on the density matrix is not established, several different proposals for the PEE that satisfy all the physical requirements have been proposed. One particular proposal we will use to calculate PEE is the *additive linear combination (ALC) proposal* [28, 29] for two-dimensional field theories, which has been shown [28, 29, 58] to satisfy all the physical requirements for generic theories.

- *The ALC proposal*: Consider an interval $A$ and partition it into three non-overlapping subregions: $A = \alpha_L \cup \alpha \cup \alpha_R$, where $\alpha$ is some subregion inside $A$ and $\alpha_L$ ($\alpha_R$) denotes the regions left (right) to it. In this configuration, the *ALC proposal* claims that:

$$s_A(\alpha) = \mathcal{I}(\alpha,\bar{A}) = \frac{1}{2}\left(S_{\alpha_L \cup \alpha} + S_{\alpha \cup \alpha_R} - S_{\alpha_L} - S_{\alpha_R}\right). \tag{8}$$

---

[5]The other four additional requirements include the *invariance under local unitary transformations, invariance under symmetry transformation, positivity* and *upper bound*.

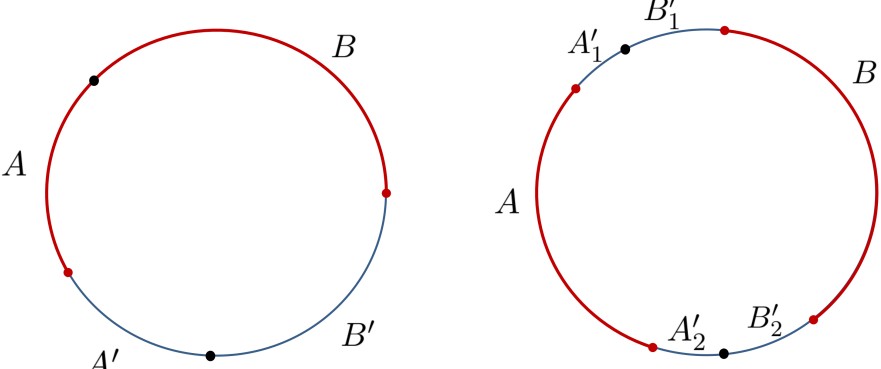

Figure 1: Figures extracted from [2], licenced under CC-BY 4.0. The purification of a region $AB$ is given by $ABB'A'$ for adjacent (left) and non-adjacent (right) intervals. When the intervals are adjacent the auxiliary system $A'B'$ consists of $A'B' = A' \cup B'$. In the non-adjacent case, the auxiliary system $A'B'$ consists of $A'B' = A'_1 \cup B'_1 \cup A'_2 \cup B'_2$.

The *ALC proposal* is supposed to be general and applicable for any theories in two dimensions, except for some disconnected systems where the order for the degrees of freedom becomes ambiguous.

Then we introduce the so-called balanced partial entanglement entropy (BPE) [1]. Consider a bipartite system $\mathcal{H}_A \otimes \mathcal{H}_B$ equipped with the density matrix $\rho_{AB}$, and a purifying system $A'B' = A' \cup B'$ which makes a pure state $|\psi\rangle$ together with $AB$, such that $\mathrm{Tr}_{A'B'} |\psi\rangle\langle\psi| = \rho_{AB}$. There exists a special partition of the auxiliary system $A'B$ such that the PEE satisfies $\mathcal{I}_{AB'} = \mathcal{I}_{A'B}$, which we call the balance condition. It represents some balance points for the correlations between $AA'$ and $BB'$.

For the cases where $A$ and $B$ are adjacent, the balance condition have the following useful and equivalent expressions

$$
\begin{aligned}
&1) s_{AA'}(A) = s_{BB'}(B); \\
&2) \mathcal{I}_{AB'} = \mathcal{I}_{A'B}; \\
&3) S_A - S_B = S_{A'} - S_{B'}.
\end{aligned} \tag{9}
$$

The above three expressions are equivalent to each other using the *ALC proposal*. For example in this case we have

$$
s_{AA'}(A) = \frac{1}{2}(S_{AA'} + S_A - S_{A'}) = \mathcal{I}_{AB} + \mathcal{I}_{AB'}. \tag{10}
$$

Compared with (8), here we have $A = \alpha$, $A' = \alpha_L$ and $\alpha_R = \emptyset$. Note that in the adjacent case we can see the PEE $\mathcal{I}_{AB}$ equals to half of the mutual information,

$$
\mathcal{I}_{AB} = s_{AA'B'}(A) = \frac{1}{2}(S_{AA'B'} + S_A - S_{A'B'}) = I(A, B)/2. \tag{11}
$$

Hereafter we will also refer to the PEE $\mathcal{I}_{AB'}$ and $\mathcal{I}_{BA'}$ as the *crossing PEEs* of the purification $|\psi\rangle$.

In the case where $A$ and $B$ are not adjacent, the balance conditions are a bit different since the complement $A'B'$ becomes disconnected. We need to further separate the disconnected regions, $A' = A'_1 \cup A'_2$ and $B' = B'_1 \cup B'_2$. As a result, they can be considered in pairs (see the right figure in Fig.1), and the balance conditions should be imposed on both pairs [1]. In other

words, we have two independent balance conditions for non-adjacent scenarios, which can be expressed as the following two equivalent ways:

$$
\begin{aligned}
&1)\ s_{AA'}(A_2) = s_{BB'}(B_2), &&s_{AA'}(A_1') = s_{BB'}(B_1'), \\
&2)\ S_{A_1'} - S_{B_1'} = S_{AA_2'} - S_{BB_2'}, &&S_{A_2'} - S_{B_2'} = S_{AA_1'} - S_{BB_1'}.
\end{aligned}
\tag{12}
$$

The condition $\mathcal{I}_{AB'} = \mathcal{I}_{A'B}$ is consistent with the above conditions.

With the balance conditions clarified, the $\mathrm{BPE}(A:B)$ is then just given by the PEE $s_{AA'}(A)$ under the balance conditions, i.e.,

$$
\mathrm{BPE}(A:B) = s_{AA'}(A)|_{\text{balance}}.
\tag{13}
$$

An interesting aspect of the BPE is that, the balance requirements also minimize the summation of the crossing PEEs.

## 2.2 $\mathrm{CFT}_2$ with gravitational anomalies and entanglement entropy

The quantum field theory we will study in this paper is the two-dimensional CFTs with gravitational anomalies [64]. Such theories have unequal left central charges $c_L$ and right central charges $c_R$ appearing in the local conformal algebras. Let us consider a covariant interval $A$ in these type of theories with endpoints given by

$$
A : (x_1, t_1) \rightarrow (x_2, t_2).
\tag{14}
$$

When the theory is in the vacuum state on a plane, the entanglement entropy for the above interval is just given by [43]

$$
S_A = \frac{c_L}{6} \log\left(\frac{z_A}{\delta}\right) + \frac{c_R}{6} \log\left(\frac{\bar{z}_A}{\delta}\right),
\tag{15}
$$

where $z_A = x - t$ and $\bar{z}_A = x + t$. The above result can be obtained using the replica trick [65–67]. In order to manifest the contribution from the anomalies, we define $z = R_A e^{i\theta_A}$, where

$$
R_A = \sqrt{x_{21}^2 - t_{21}^2}, \qquad x_{21} = x_2 - x_1, \qquad t_{21} = t_2 - t_1
\tag{16}
$$

is the proper length of the interval $A$ and $\theta_A$ is the boost angle of $A$ in spacetime. It will be more convenient to use the boost angle $\kappa_A = -i\theta_A$, such that

$$
\kappa_A = \tanh^{-1}\left(\frac{t_{21}}{x_{21}}\right).
\tag{17}
$$

Then in terms of $R_A$ and $\kappa_A$ the entanglement entropy can be written as

$$
S_A = \frac{c_L + c_R}{6} \log\left(\frac{R_A}{\delta}\right) - \frac{c_L - c_R}{6} \kappa_A,
\tag{18}
$$

where the second term is exactly the contribution from anomalies. Also, the entanglement entropy for a static interval with length $R_A$ at finite temperature is given in [43],

$$
S_A = \frac{c_R + c_L}{12} \log\left(\frac{\beta_L \beta_R}{\pi^2 \delta^2} \sinh\left(\frac{\pi R_A}{\beta_L}\right) \sinh\left(\frac{\pi R_A}{\beta_R}\right)\right) + \frac{c_R - c_L}{12} \log\left(\frac{\sinh\left(\frac{\pi R_A}{\beta_R}\right) \beta_R}{\sinh\left(\frac{\pi R_A}{\beta_L}\right) \beta_L}\right).
\tag{19}
$$

In the rest of this paper, in order to distinguish the anomalous and non-anomalous contribution we define

$$c_{ano} \equiv (c_L - c_R), \qquad c_{geo} \equiv (c_L + c_R). \tag{20}$$

We call the non-anomalous contribution *geometric* since it corresponds to pure geometric quantities in holography. The entanglement entropy is therefore decomposed into two terms, which are proportional to $c_{geo}$ and $c_{ano}$ respectively, i.e.

$$S_A = S_A^{geo} + S_A^{ano}. \tag{21}$$

Similar decomposition will be applied to other quantities like the BPE, reflected entropy and the quantity associated to the EWCS on gravity side.

## 3 Covariant balanced partial entanglement and gravitational anomalies

In this section we will calculate the BPE for CFT$_2$ with or without gravitational anomaly for generic covariant configurations, and compare the results with the reflected entropy carried out in [44]. Note that, the BPE can be defined in non-holographic systems, hence the calculations in this section goes beyond AdS/CFT.

### 3.1 Balanced partial entanglement for adjacent intervals

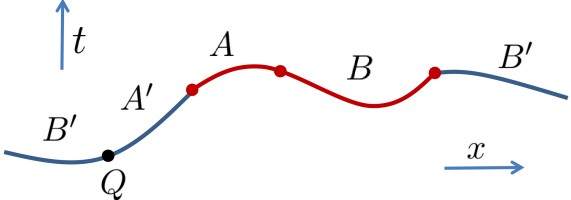

Figure 2: Setup for two adjacent covariant intervals $A$ and $B$ in the vacuum state of a CFT$_2$ with gravitational anomalies. The auxiliary system $A'B'$ is the complement of $AB$ on the spatial curve, and is partitioned by the point $Q$.

Now we consider two adjacent covariant intervals $A$ and $B$ in the vacuum state of a CFT$_2$ with gravitational anomalies, as illustrated in Fig.2. The intervals are given by

$$A : (x_1, t_1) \to (x_2, t_2), \qquad B : (x_2, t_2) \to (x_3, t_3), \tag{22}$$

which are embedded in an infinitely long and spacelike curve. The system settled on the curve is in a pure state $|\Psi\rangle$ and the complement $A'B'$ can be considered as the purifying system for the mixed state represented by the reduced matrix $\rho_{AB}$ on $AB$. Before imposing the balance conditions, the partition point $Q$ which divides $A'B'$ into $A' \cup B'$ can be any point spacelike-separated from $(x_1, t_1)$ and $(x_3, t_3)$ outside the causal development of $AB$. If we consider a CFT without gravitational anomalies, the solutions for the only balance condition (9) form a curve in spacetime, which will give us different $BPE(A, B)$ when moving the partition point $Q$ along this curve. In other words, the balance conditions cannot uniquely determine the partition points $Q$ hence give different BPEs in the case of the CFT without gravitational anomalies.

This contradicts with the claim of [2] that the BPE is an intrinsic measure of the mixed state correlation hence should be unique when $\rho_{AB}$ is settled down. In the following we will show that when the contribution from gravitational anomalies is considered, the partition point $Q$ will be uniquely determined and give the right $BPE(A,B)$ that reproduces the reflected entropy.

If one considers $c_L$ and $c_R$ as independent, then the balance conditions should be imposed independently to both the logarithmic term and the term from gravitational anomalies. In such a way the balance condition (9) splits into two independent constraints,

$$S_A^{geo} - S_B^{geo} = S_{A'}^{geo} - S_{B'}^{geo}, \qquad S_A^{ano} - S_B^{ano} = S_{A'}^{ano} - S_{B'}^{ano}. \tag{23}$$

A similar strategy was previously used to solve the covariant BPE [68] in the 3-dimensional flat holography [20,69,70]. Let us denote the position of $Q$ as $(x_0, t_0)$, and set $(x_1, t_1) = (0, 0)$ without loss of generality. However, the solutions to the above balance conditions in terms of the parameters $x_{2,3}$ and $t_{2,3}$ are quite complicated. In the following we will use the other set of parameters we defined in section 2.2,

$$\begin{aligned} x_2 &= R_A \cosh[\kappa_A], & x_3 &= R_{AB} \cosh[\kappa_{AB}], \\ t_2 &= R_A \sinh[\kappa_A], & t_3 &= R_{AB} \sinh[\kappa_{AB}]. \end{aligned} \tag{24}$$

The proper length $R_B$ and boost angle $\kappa_B$ are determined by

$$\tanh[\kappa_B] = \frac{R_A \sinh[\kappa_A] - R_{AB} \sinh[\kappa_{AB}]}{R_A \cosh[\kappa_A] - R_{AB} \cosh[\kappa_{AB}]}, \quad R_B^2 = R_A^2 + R_{AB}^2 - 2R_A R_{AB} \cosh[\kappa_A - \kappa_{AB}], \tag{25}$$

then the solutions to the balance conditions are given by

$$\begin{aligned} x_0 &= \frac{R_A R_{AB}(2R_A \cosh(\kappa_{AB}) - R_{AB} \cosh(\kappa_A))}{-4R_A R_{AB} \cosh(\kappa_A - \kappa_{AB}) + 4R_A^2 + R_{AB}^2}, \\ t_0 &= \frac{R_A R_{AB}(2R_A \sinh(\kappa_{AB}) - R_{AB} \sinh(\kappa_A))}{-4R_A R_{AB} \cosh(\kappa_A - \kappa_{AB}) + 4R_A^2 + R_{AB}^2}. \end{aligned} \tag{26}$$

When the balance conditions determine the partition point $Q$, which we refer to as the balance point hereafter, the entanglement entropies for all the subsets on the total system can be calculated by the formula (18). The BPE$(A,B)$ is then given by the PEE $s_{AA'}(A)$ satisfying the balance conditions,

$$\begin{aligned} \text{BPE}(A,B) = s_{AA'}(A) &= \frac{1}{2}(S_{AA'} + S_A - S_{A'})|_{balanced} \\ &= \frac{c_{geo}}{12} \log\left(\frac{R_A R_B}{\delta R_{AB}}\right) - \frac{c_{ano}}{12}(\kappa_A + \kappa_B - \kappa_{AB}) + \frac{c_{geo}}{12} \log(2), \end{aligned} \tag{27}$$

which exactly matches with the reflected entropy calculated in [44].

## 3.2 Minimizing the crossing correlations with the balance conditions

In the adjacent case the PEE $\mathcal{I}(A,B) = I(A,B)/2$, which is just half the mutual information, then

$$s_{AA'}(A) = I(A,B)/2 + \mathcal{I}(A,B'). \tag{28}$$

We use the second expression of (9): $\mathcal{I}_{AB'} = \mathcal{I}_{A'B}$. In [2], it was shown that in the static configurations with no gravitational anomalies, the balance conditions indeed minimize the crossing PEEs. More explicitly we have

$$(\mathcal{I}_{AB'} + \mathcal{I}_{BA'})|_{balance} = 2\mathcal{I}_{AB'}|_{balance} = (\mathcal{I}_{AB'} + \mathcal{I}_{BA'})|_{minimized}, \tag{29}$$

where the minimization is among all the partition of $A'B'$. Furthermore, the minimized crossing PEE is given by a constant

$$\mathcal{I}_{AB'}|_{balance} = (\mathcal{I}_{AB'} + \mathcal{I}_{BA'})/2|_{minimized} = \frac{c}{6}\log 2. \tag{30}$$

This constant is a generalization of the so-called Markov gap [71], which is defined as the difference between the reflected entropy and the mutual information. It was proposed in [2] that the crossing PEE at the balance condition is a universal tripartite correlation that is independent from purifications. This proposal has been tested in many configurations including both the holographic and non-holographic models [2]. This is necessary for the claim that the BPE is an intrinsic measure for the mixed state correlations.

Here we explore the minimized crossing PEE in covariant configurations with gravitational anomalies. Firstly, we take a look at the crossing PEE under the balance conditions. It is easy to see that, the first term plus the second term in the BPE$(A, B)$ (27) give the half the mutual information $I(A, B)/2 = (S_A + S_B - S_{AB})/2$. Then we find that the difference between the BPE$(A, B)$ and $I(A, B)/2$ is just the third term in (27),

$$\begin{aligned}
\mathcal{I}_{AB'}|_{balance} &= s_{AA'}(A)|_{balance} - \mathcal{I}_{AB} \\
&= \text{BPE}(A, B) - I(A, B)/2 \\
&= \frac{c_{geo}}{12}\log(2),
\end{aligned} \tag{31}$$

which is a constant and does not receive contribution from the anomalies.

Let us go away from the balance point for a moment and consider undetermined $(x_0, t_0)$ outside $AB$. The PEEs like $\mathcal{I}_{AB'}$ can be calculated using the *ALC* proposal (8), hence we can compute the crossing PEE and decompose it into two contributions:

$$\mathcal{I}_C \equiv (\mathcal{I}_{AB'} + \mathcal{I}_{BA'}) = \mathcal{I}_C^{geo} + \mathcal{I}_C^{ano}, \tag{32}$$

where $\mathcal{I}_C^{geo} \propto c_{geo}$ and $\mathcal{I}_C^{ano} \propto c_{ano}$. These two contributions are functions of $x_0$ and $t_0$ and are given by

$$\mathcal{I}_C^{geo} = \frac{c_{geo}}{24}\log\left[\frac{R_{AB}^4\left(2R_A t_0 \sinh(\kappa_A) - 2R_A x_0 \cosh(\kappa_A) + R_A^2 - t_0^2 + x_0^2\right)^2}{R_A^2 R_B^2\left(x_0^2 - t_0^2\right)\left(2R_{AB} t_0 \sinh(\kappa_{AB}) - 2R_{AB} x_0 \cosh(\kappa_{AB}) + R_{AB}^2 - t_0^2 + x_0^2\right)}\right],$$

$$\begin{aligned}
\mathcal{I}_C^{ano} = \frac{c_{ano}}{24}\left(\log\left[\frac{(t_0 + x_0)\left(e^{-\kappa_A}R_A + t_0 - x_0\right)^2\left(-e^{\kappa_{AB}}R_{AB} + t_0 + x_0\right)}{(t_0 - x_0)\left(-e^{\kappa_A}R_A + t_0 + x_0\right)^2\left(e^{-\kappa_{AB}}R_{AB} + t_0 - x_0\right)}\right]\right) \\
+ \frac{c_{ano}}{12}(\kappa_A - 2\kappa_{AB} + \kappa_B),
\end{aligned} \tag{33}$$

where we used the parameters defined in (24). As expected, one can check that the crossing PEE under the balance condition (26) is just given by

$$\mathcal{I}_C|_{balance} = \frac{c_{geo}}{6}\log 2. \tag{34}$$

Now we explore whether the crossing PEE reaches its minimal value under the balance conditions. One can explicitly verify that when the balance conditions (26) are satisfied, we have

$$\begin{aligned}
\frac{\partial \mathcal{I}_C^{geo}}{\partial x_0}\Big|_{balance} = 0, \quad \frac{\partial \mathcal{I}_C^{geo}}{\partial t_0}\Big|_{balance} = 0, \\
\frac{\partial \mathcal{I}_C^{ano}}{\partial x_0}\Big|_{balance} = 0, \quad \frac{\partial \mathcal{I}_C^{ano}}{\partial t_0}\Big|_{balance} = 0,
\end{aligned} \tag{35}$$

which means the two contributions $\mathcal{I}_C^{geo}$ and $\mathcal{I}_C^{ano}$, as well as their summation $\mathcal{I}_C$, reaches a saddle point at the balance point. In the static case there is a general argument demonstrating that the balance point is the minimum point of $\mathcal{I}_C$. Here we directly calculate the second order derivatives of $\mathcal{I}_C$ to check whether the saddle is a minimum point. The criterion for the saddle to be a minimum point is to satisfy the following two inequalities

$$\frac{\partial^2 \mathcal{I}_C}{\partial^2 x_0}\bigg|_{balance} \frac{\partial^2 \mathcal{I}_C}{\partial^2 t_0}\bigg|_{balance} - \left(\frac{\partial^2 \mathcal{I}_C}{\partial x_0 \partial t_0}\right)^2 \bigg|_{balance} > 0 , \tag{36}$$

$$\frac{\partial^2 \mathcal{I}_C}{\partial^2 x_0}\bigg|_{balance} > 0 . \tag{37}$$

It is indeed hard to explicitly check the above inequalities in general as the expression become very complicated. Nevertheless, we find that by using (24), the left-hand side of (36) is always some square function multiplying $c_{geo}^2 - c_{ano}^2$. For example when we fix $\kappa_A = 0$ we find

$$\frac{\partial^2 \mathcal{I}_C}{\partial^2 x_0}\bigg|_{balance} \frac{\partial^2 \mathcal{I}_C}{\partial^2 t_0}\bigg|_{balance} - \left(\frac{\partial^2 \mathcal{I}_C}{\partial x_0 \partial t_0}\right)^2 \bigg|_{balance}$$
$$= \frac{\left(c_{geo}^2 - c_{ano}^2\right)\left(4R_A^2 + R_{AB}^2 - 4R_A R_{AB} \cosh(\kappa_{AB})\right)^4}{576 R_A^4 R_{AB}^4 \left(R_A^2 + R_{AB}^2 - 2R_A R_{AB} \cosh(\kappa_{AB})\right)^2} . \tag{38}$$

Since $c_{geo} > c_{ano}$, the inequality (36) holds in general. For the second criterion we calculate the left-hand side of (37), which also has complicated expression. We find that it can be written in the following formula

$$\frac{\partial^2 \mathcal{I}_C}{\partial^2 t_0}\bigg|_{balance} = \frac{\partial^2 \mathcal{I}_C}{\partial^2 x_0}\bigg|_{balance} = \mathcal{N}_1 c_{geo} + \mathcal{N}_2 c_{ano} , \tag{39}$$

where $\mathcal{N}_{1,2}$ are two functions of the four parameters that determines $A$ and $B$. They satisfy the following equations

$$\mathcal{N}_1 + \mathcal{N}_2 = \frac{(e^{\kappa_A} R_{AB} - 2e^{\kappa_{AB}} R_A)^4}{24 R_A^2 R_{AB}^2 (e^{\kappa_{AB}} R_A - e^{\kappa_A} R_{AB})^2} > 0 ,$$
$$\mathcal{N}_1 - \mathcal{N}_2 = \frac{e^{-2(\kappa_A + \kappa_{AB})} (e^{\kappa_{AB}} R_{AB} - 2e^{\kappa_A} R_A)^4}{24 R_A^2 R_{AB}^2 (e^{\kappa_A} R_A - e^{\kappa_{AB}} R_{AB})^2} > 0 , \tag{40}$$

which are enough to demonstrate that the inequality (37) is satisfied in general. One can furthermore verify that the balance point is a minimum point for $\mathcal{I}_C^{geo}$, but is neither a minimum or maximal point for $\mathcal{I}_C^{ano}$.

In summary we demonstrated that, in general covariant scenarios where $A$ and $B$ are adjacent, the crossing PEE reaches its minimal value under the balance condition, which is a constant conjectured to be universal,

$$\mathcal{I}_C|_{balance} = 2\mathcal{I}_{AB'}|_{balance} = \mathcal{I}_C|_{minimized} = c_{geo}/6 \log 2 . \tag{41}$$

Since there is no contribution proportional to $c_{ano}$, we conclude that this universal tripartite entanglement has no contribution from the gravitational anomalies. The BPE in these cases can be written as

$$\text{BPE}(A,B) = \frac{1}{2} I(A,B) + \frac{c_{geo}}{12} \log 2 . \tag{42}$$

In the covariant cases without gravitational anomalies, where $c_L = c_R = c$, we found the second term become $c/6 \log 2$, which is exactly the universal constant we got in [2]. We therefore have checked the universality of the minimized crossing PEE in the covariant scenarios with gravitational anomalies.

### 3.3 Balanced partial entanglement for non-adjacent intervals

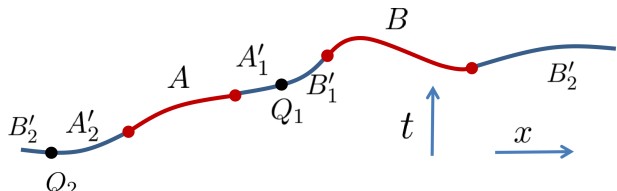

Figure 3: Figure extracted from [2], licenced under CC-BY 4.0. Setup for two non-adjacent covariant intervals $A$ and $B$ in the vacuum state of a $CFT_2$ with gravitational anomalies. The auxiliary system $A'B'$ is partitioned by the points $Q_1, Q_2$ into four subintervals $A'_1, A'_2, B'_1, B'_2$.

Here we consider the scenarios where $A$ and $B$ are spacelike-separated regions with the following endpoints

$$A: (x_1, t_1) \rightarrow (x_2, t_2), \qquad B: (x_4, t_4) \rightarrow (x_5, t_5). \tag{43}$$

The purification $AA'BB'$ is again the vacuum state on an infinitely long line. The purifying system $A'B'$ is now disconnected and will be divided into four regions by two partition points $Q_1$ and $Q_2$, see Fig.3. We denote the coordinates for the partition points to be

$$Q_2: (x_0, t_0), \qquad Q_1: (x_3, t_3), \tag{44}$$

such that the subintervals in the purifying system are given by

$$\begin{aligned}
&A'_1: (x_2, t_2) \rightarrow (x_3, t_3), \quad B'_1: (x_3, t_3) \rightarrow (x_4, t_4),\\
&A'_2: (x_0, t_0) \rightarrow (x_1, t_1),\\
&B'_2: -\infty \rightarrow (x_0, t_0) \cup (x_5, t_5) \rightarrow \infty.
\end{aligned} \tag{45}$$

Similarly, we divide the entanglement entropy into contributions from the *geometric* terms and the gravitational anomalies. In this case the balance conditions (12) turns out to be four independent conditions

$$S^{geo}_{A'_1} - S^{geo}_{B'_1} = S^{geo}_{AA'_2} - S^{geo}_{BB'_2}, \quad S^{geo}_{A'_2} - S^{geo}_{B'_2} = S^{geo}_{AA'_1} - S^{geo}_{BB'_1}, \tag{46}$$

$$S^{ano}_{A'_1} - S^{ano}_{B'_1} = S^{ano}_{AA'_2} - S^{ano}_{BB'_2}, \quad S^{ano}_{A'_2} - S^{ano}_{B'_2} = S^{ano}_{AA'_1} - S^{ano}_{BB'_1}, \tag{47}$$

which are enough to determine the positions of the two partition points.

The position of $Q_2$ will be on the left of $A$ as long as $A$ have a smaller size than that of $B$. Otherwise, it will be on the right-hand side of $B$. Let us assume that $A$ has a smaller size, then the coordinates of all the points need to satisfy certain constrains. Firstly, both $A$ and $B$ should be spacelike intervals, and the whole system should be settled on a spacelike curve. Secondly, we should have,

$$x_0 < x_1 < x_2 < x_3 < x_4 < x_5. \tag{48}$$

The above inequality will help us select the proper solutions to the balance conditions. For the non-adjacent cases it will be convenient to introduce the follow cross ratios

$$\eta = \frac{(t_1 - t_2 - x_1 + x_2)(t_4 - t_5 - x_4 + x_5)}{(t_1 - t_4 - x_1 + x_4)(t_2 - t_5 - x_2 + x_5)}, \quad \bar{\eta} = \frac{(t_1 - t_2 + x_1 - x_2)(t_4 - t_5 + x_4 - x_5)}{(t_1 - t_4 + x_1 - x_4)(t_2 - t_5 + x_2 - x_5)}. \tag{49}$$

We can again set $x_1 = t_1 = 0$ without losing generality. At first. we solve the equation (47) and get

$$x_0 = \frac{2t_3^2 x_5 - t_5^2 x_3 + x_3 x_5 (x_5 - 2x_3)}{(-2t_3 + t_5 + 2x_3 - x_5)(-2t_3 + t_5 - 2x_3 + x_5)},$$

$$t_0 = -\frac{-2t_3^2 t_5 + t_3(t_5 - x_5)(t_5 + x_5) + 2t_5 x_3^2}{(-2t_3 + t_5 + 2x_3 - x_5)(-2t_3 + t_5 - 2x_3 + x_5)}. \tag{50}$$

Then we plug the above solution into (47) to solve $(x_3, t_3)$. However, it is quite hard to get the full analytic solution with all the other three endpoints of $AB$ free. We need to fix two endpoints, for example $(x_2, t_2)$ and $(x_4, t_4)$, hence only the two parameters $(x_5, t_5)$ are free. It will further simplify the calculation by replacing $(x_5, t_5)$ with $(\eta, \bar{\eta})$. For example, if we set

$$\textbf{case 1}: \quad t_2 = 0, x_2 = 1, t_4 = 0, x_4 = 3, \tag{51}$$

we find the solution for (47)

$$x_3 = \frac{\sqrt{\eta}(9\sqrt{\bar{\eta}} + 6) + 6\sqrt{\bar{\eta}} + 3}{(3\sqrt{\eta} + 1)(3\sqrt{\bar{\eta}} + 1)}, \quad t_3 = \frac{3(\sqrt{\eta} - \sqrt{\bar{\eta}})}{(3\sqrt{\eta} + 1)(3\sqrt{\bar{\eta}} + 1)}. \tag{52}$$

Plugging the above solutions (50) and (52) into the PEE

$$s_{AA'}(A) = \frac{1}{2}\left(S_{AA_1'} + S_{AA_2'} - S_{A_1'} - S_{A_2'}\right), \tag{53}$$

then the BPE is given by $s_{AA'}(A)|_{balance}$:

$$\text{BPE}(A, B) = \frac{c_{geo}}{24} \log \frac{(\sqrt{\eta} + 1)(\sqrt{\bar{\eta}} + 1)}{(\sqrt{\eta} - 1)(\sqrt{\bar{\eta}} - 1)} + \frac{c_{ano}}{24} \log \frac{(\sqrt{\eta} + 1)(\sqrt{\bar{\eta}} - 1)}{(\sqrt{\eta} - 1)(\sqrt{\bar{\eta}} + 1)}, \tag{54}$$

which is exactly the same as the reflected entropy calculated in [44]. Although it is hard for us to get the analytic BPE for all the endpoints of $AB$ settled to be free, we can adjust the position of the points $(t_2, x_2)$ and $(t_4, x_4)$. For example, we can consider the following choices,

$$\textbf{case 2}: t_2 = \frac{1}{2}, x_2 = 1, t_4 = 1, x_4 = 3,$$

$$\textbf{case 3}: t_2 = \frac{1}{2}, x_2 = 1, t_4 = -\frac{1}{2}, x_4 = 5. \tag{55}$$

We find that, no matter how we adjust the other two endpoints, the BPE$(A, B)$ will always be given by the same formula (54) as long as they are settled on a spacelike curve. Hereafter we will refer to the terms proportional to $c_{geo}, c_{ano}$ of the BPE, or the reflected entropy, as the *geometric* term and the *anomalous* term respectively. One may also test (numerically) that the crossing PEE $\mathcal{I}_C \equiv \mathcal{I}(A, B_1' \cup B_2') + \mathcal{I}(B, A_1' \cup A_2')$ reaches its minimal value at the balance point. In Fig. 4, we illustrate one example with a set of parameters provided.

# 4 Correction to entanglement wedge cross-section from gravitational anomalies

## 4.1 Correction to Ryu-Takayanagi surfaces from gravitational anomalies

Now we turn to TMG [72, 73] which are proposed to be dual to the holographic CFTs with gravitational anomalies. The action of TMG in AdS includes the Einstein-Hilbert term, a cosmological constant term and an additional CS term,

$$\mathcal{S}_{\text{TMG}} = \frac{1}{16\pi G} \int d^3x \; \sqrt{-g}\left[R + \frac{2}{\ell^2} + \frac{1}{2\mu}\varepsilon^{\alpha\beta\gamma}\left(\Gamma_{\alpha\sigma}^{\rho}\partial_{\beta}\Gamma_{\gamma\rho}^{\sigma} + \frac{2}{3}\Gamma_{\alpha\sigma}^{\rho}\Gamma_{\beta\eta}^{\sigma}\Gamma_{\gamma\rho}^{\eta}\right)\right]. \tag{56}$$

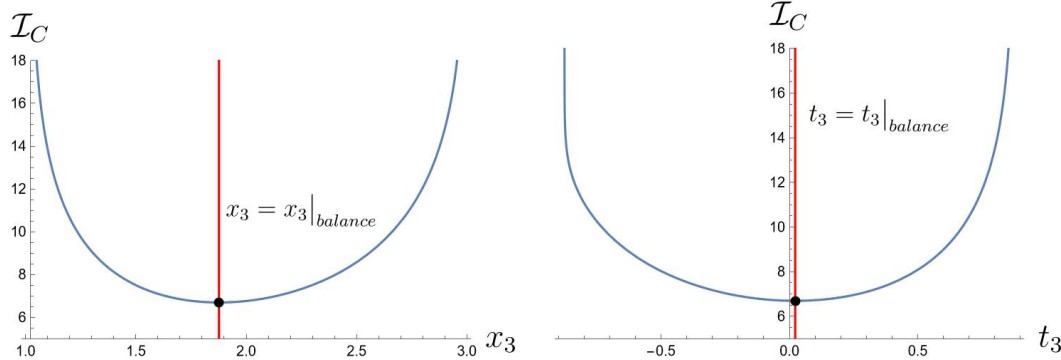

Figure 4: We evaluate case 1 (51) with $\eta = 1/5, \bar{\eta} = 1/6, c_{geo} = 55, c_{ano} = 37$, and we additionally consider that the first balance requirement (50) is satisfied. In this case, the crossing PEE $\mathcal{I}_C$ is a function of $x_3, t_3$. We will let either $x_3$ or $t_3$ satisfy the second balance requirement (52) and see whether the saddle of $\mathcal{I}_C$ is settled at the balance point of the other parameter. Left figure: The blue curve describes $\mathcal{I}_C = \mathcal{I}_C(x_3, t_3|_{balance})$ and the red vertical line corresponds to $x_3$ at the balance point. Right figure: The blue curve describes $\mathcal{I}_C = \mathcal{I}_C(x_3|_{balance}, t_3)$ and the red vertical line corresponds to $t_3$ at the balance point. The black dot in the two figures denotes the same balanced crossing PEE which is indeed the minimum point. Other cases can be verified in a similar manner.

The Einstein gravity is recovered in the limit $\mu \to \infty$. Specializing to the locally $\mathrm{AdS}_3$ solutions to the gravity, the dual CFT is described by Virasoro algebra with left and right central charges [74]. In this paper, we adopt the notations from [43]:

$$c_L = \frac{3\ell}{2G}(1 - \frac{1}{\mu\ell}), \qquad c_R = \frac{3\ell}{2G}(1 + \frac{1}{\mu\ell}). \tag{57}$$

In the following we set $\ell = 1$. Accordingly the correspondence between the two-dimensional CFTs with gravitational anomalies and the locally $\mathrm{AdS}_3$ solutions to TMG is established [75–78]. The black hole entropy in such theories is derived in [48] by extending the Wald Formula [45–47]. The result shows that the correction from the CS term is proportional to the area of the inner horizon. This is also consistent with earlier explorations on this topic via some indirect methods [75, 79–81].

Interestingly, the holographic entanglement entropy in this context was also explicitly calculated in [43] by generalizing the Lewkowycz-Maldacena prescription [82]. The new picture broadens the RT geodesic into a worldline action for a spinning particle in three-dimensional curved spacetime,

$$S_{EE} = \frac{1}{4G} \int_C d\tau \left( \sqrt{g_{\alpha\beta} \dot{X}^{\alpha} \dot{X}^{\beta}} + \frac{1}{\mu} \tilde{n} \cdot \nabla n \right), \tag{58}$$

where $C$ represents the worldline of the particle, $\tilde{n}$ and $n$ are unit spacelike and timelike vectors along the worldline and are normal to each other, $v$ is the tangent vector along the worldline $C$, and $\nabla n \equiv v^{\mu} \nabla_{\mu} n$. More explicitly they satisfy the following constraints,

$$n \cdot v = 0, \quad \tilde{n} \cdot v = 0, \quad n \cdot \tilde{n} = 0, \quad n^2 = -1, \quad \tilde{n}^2 = 1. \tag{59}$$

We will limit our discussion on the locally $\mathrm{AdS}_3$ spacetime, where the worldline that extremizes the action functional turns out to be a geodesic, then the first term in (58) is just the familiar Ryu-Takayanagi term which is contributed from the Einstein-Hilbert action while the

second term is the contribution from the CS term. It has been verified that the holographic entanglement entropy (58) exactly matches with the CFT results (18) computed by the replica technique provided (57). One may consult [43, 83, 84] for more discussions about the above entanglement entropy functional.

Here we talk about a subtlety in the derivation of (58) which will be important in our later discussion for the EWCS. The subtlety is originated from the fact that the action of TMG is not manifestly invariant under diffeomorphism. According to [43], if we give the following metric ansatz,

$$ds^2 = e^{\epsilon\phi(\sigma)}\delta_{ab}d\sigma^a d\sigma^b + (g_{yy} + K_a\sigma^a + \cdots)dy^2 + e^{\epsilon\phi(\sigma)}U_a(\sigma, y)d\sigma^a dy \tag{60}$$

to the near cone region when applying the replica trick in the bulk, the contribution from the CS term to the entropy is given by

$$S_{CS} = \frac{1}{16\mu G}\int_C dy\,\epsilon^{ab}\partial_a U_b\,. \tag{61}$$

In the above parameterization, $y$ is the coordinate along the geodesic, $\sigma_{1,2}$ are the perpendicular coordinates, $K_a$ is the extrinsic curvatures, and $\phi(\sigma)$ stores the information of the regularization of the cone. The second term of (58) is indeed another formula to rewrite (61), and the two normal vectors can be chosen to be

$$n = \frac{\partial}{\partial\sigma^1}, \quad \tilde{n} = \frac{\partial}{\partial\sigma^2}, \tag{62}$$

which depend on the choice of coordinates hence are not covariant. More explicitly if we perform an infinitesimal $y$-dependent rotation for the perpendicular coordinates $\delta\sigma^a = -\theta(y)\epsilon^a{}_b\sigma^b$, the variation of the integrand in (61) is just a total derivative,

$$\delta\left(\sigma^{ab}\partial_a U_b(y)\right) = 4\theta'(y)\,. \tag{63}$$

Interestingly, the entropy contributed from the CS term shifts by a boundary term from the endpoints of the integral [43],

$$\delta S_{CS} = \frac{1}{4G\mu}\left(\theta(y_f) - \theta(y_i)\right), \tag{64}$$

where $y_f$ and $y_i$ in this case represent the two points where the geodesic anchored on the boundary.

If the contribution from the CS term is coordinate dependent, how can it capture any intrinsic physical information? Authors in [43] proposed that, this contribution should capture the intrinsic information of how the local coordinates twist along the geodesic. Fortunately, since the integration only depend on the boundary terms, only the difference between the initial and the final status of the normal vector $n$ (or $\tilde{n}$) matters. The integral will make sense if there is a physical way to determine $n_f$ given $n_i$. This is true as the near curve metric at the boundary points $y_i$ and $y_f$ should coincide with the spacetime we used to define the boundary CFT. If we take $n_i \propto \partial_t$, then (62) instructs that

$$n_f \propto \partial_t\,, \tag{65}$$

where $t$ is the time coordinate of the boundary CFT. With $n_i$ chosen, the CS term contribution can be captured by the following formula [43],

$$S_{CS} = \frac{1}{4G\mu}\log\left(\frac{q(s_f)\cdot n_f - \tilde{q}(s_f)\cdot n_f}{q(s_i)\cdot n_i - \tilde{q}(s_i)\cdot n_i}\right), \tag{66}$$

where $s$ parametrizes the geodesic and the two vectors $q(s)$ and $\tilde{q}(s)$ give a reference parallel transported normal frame satisfying

$$
\begin{aligned}
& q^2 = -1, \quad \tilde{q}^2 = 1, \quad q \cdot v = 0, \quad \tilde{q} \cdot v = 0, \\
& q \cdot \tilde{q} = 0, \quad \nabla q^\nu = 0, \quad \nabla \tilde{q}^\nu = 0.
\end{aligned}
\tag{67}
$$

It has been verified in [43] that the above formula (66) exactly captures the contribution from the gravitational anomalies to the entanglement entropy on the field theory side. Also (66) can be applied to the black hole horizon which is a closed loop. Although there is no boundary condition for the normal vector $n$ in this case, the integration along the loop will give the same answer as [48] for any choice of $n$.

To conclude, the CS term contribution associated to a closed black hole horizon, or the RT surface anchored on the boundary, can get rid of the unphysical dependence on the choice of coordinates. The main reason is that, the change of the integral (61) under a change of the coordinates is up to a boundary term, which can be physically determined by the boundary conditions of the bulk metric. For the case of the EWCS, it is natural to consider correction from the CS term to be the integration on the EWCS instead of the whole RT surface. A naive and straightforward guess for the generalized EWCS in TMG could be[6],

$$
\begin{aligned}
E_W(A, B) &= E_W^{EH}(A, B) + E_W^{CS}(A, B), \\
E_W^{EH}(A, B) &= \frac{1}{4G} Length(\Sigma_{AB}), \\
E_W^{CS}(A, B) &= \frac{1}{4G\mu} \int_{\Sigma_{AB}} d\tau \, \tilde{n} \cdot \nabla n,
\end{aligned}
\tag{68}
$$

where $\Sigma_{AB}$ is the cross section, the first term is the familiar contribution from the Einstein-Hilbert action while the second term represents the correction from the CS term. Given the duality between TMG and the $CFT_2$ with gravitational anomalies, it is natural to conjecture that the measure $E_W(A, B)$ is the holographic dual of the BPE or the reflected entropy in the boundary field theory as in the cases of no gravitational anomalies,

$$
E_W(A, B) = BPE(A, B) = S_R(A, B). \tag{69}
$$

More explicitly, the first term $E_W^{EH}(A, B)$ in (68) should reproduce the *geometric* term of the BPE and the second term $E_W^{CS}(A, B)$ should reproduce the *anomalous* term. In the following we will analytically verify that $E_W^{EH}(A, B)$ indeed gives the same result as the *geometric* term of the BPE for generic covariant configurations. We will then give a new geometric prescription to calculate $E_W^{CS}(A, B)$, and check that the correction reproduces the *anomalous* term of the BPE.

## 4.2 Length of the EWCS in covariant scenarios

In the static cases, the EWCS $\Sigma_{AB}$ is defined as the minimal cross-section of the entanglement wedge $\mathcal{W}_{AB}$ which separates $A$ from $B$. In covariant scenarios, the entanglement wedge no longer settle on a constant time slice such that the minimal surface for the EWCS could be generalized to the extremal surface, similar to the case that the RT formula is generalized to the HRT formula. More explicitly the EWCS will be the saddle geodesic among all the geodesics anchored on different pieces of $\mathcal{E}_{AB}$.

---

[6]See [44] for some earlier discussion.

### 4.2.1 Poincaré coordinates vs light-cone coordinates

Before we move on to explicit calculations, we make clear what coordinate systems we will extensively use and list some useful formulas to simplify calculations. In this paper we will focus on Poincaré $AdS_3$ with the metric given by

$$ds^2 = \frac{-dt^2 + dx^2 + dz^2}{z^2} \, . \tag{70}$$

It is useful to implement the transformations

$$U = \frac{t+x}{2}, \quad V = \frac{x-t}{2}, \quad \rho = \frac{2}{z^2}, \tag{71}$$

to work in light-cone coordinates:

$$ds^2 = \frac{d\rho^2}{4\rho^2} + 2\rho \, dU dV \, . \tag{72}$$

Accordingly the points $(t_i, x_i)$ in the previous discussion will be denoted by $(U_i, V_i)$ in this section.

In light-cone coordinates, any two spacelike-separated points, e.g. $(U_1, V_1, \rho_1)$ and $(U_2, V_2, \rho_2)$, can be connected by a geodesic chord parametrized by

$$U = \frac{l_U}{2} \tanh \tau + c_U, \quad V = \frac{l_V}{2} \tanh \tau + c_V, \quad \rho = \frac{2\cosh^2 \tau}{l_U l_V}, \tag{73}$$

where $\tau$ is an affine parameter such that the tangent vector along the curve is given by

$$v = \frac{1}{l_V \rho} \partial_U + \frac{1}{l_U \rho} \partial_V + \frac{4\rho (U - c_U)}{l_U} \partial_\rho \, . \tag{74}$$

The length of this geodesic chord is given by a general formula derived in the appendix C in [21]:

$$L_{\text{AdS}}(U_1, V_1, \rho_1, U_2, V_2, \rho_2) =$$
$$\frac{1}{2} \log \left[ \frac{\rho_2 (\rho_2 + X) + \rho_1 (\rho_2 Y (2\rho_2 + X) + X) + (\rho_1 + \rho_2 \rho_1 Y)^2}{2\rho_1 \rho_2} \right], \tag{75}$$

where

$$\begin{aligned} Y &= 2(U_1 - U_2)(V_1 - V_2) \, , \\ X &= \sqrt{\rho_1^2 + 2\rho_2 \rho_1 (\rho_1 Y - 1) + (\rho_2 + \rho_1 \rho_2 Y)^2} \, . \end{aligned} \tag{76}$$

For parametrization (73), the reference normal frame $q, \tilde{q}$ determined by (67) can be chosen to be

$$\begin{aligned} q &= \frac{l_U}{\sqrt{2l_U l_V \rho}} \partial_U - \frac{l_V}{\sqrt{2l_U l_V \rho}} \partial_V \, , \\ \tilde{q} &= \sqrt{\frac{2}{l_U l_V \rho}} \left( -\frac{l_U (U + V - c_U - c_V)}{l_U + l_V} \partial_U - \frac{l_V (U + V - c_U - c_V)}{l_U + l_V} \partial_V + 2\rho \partial_\rho \right) , \end{aligned} \tag{77}$$

up to an overall sign related to a choice of handedness.

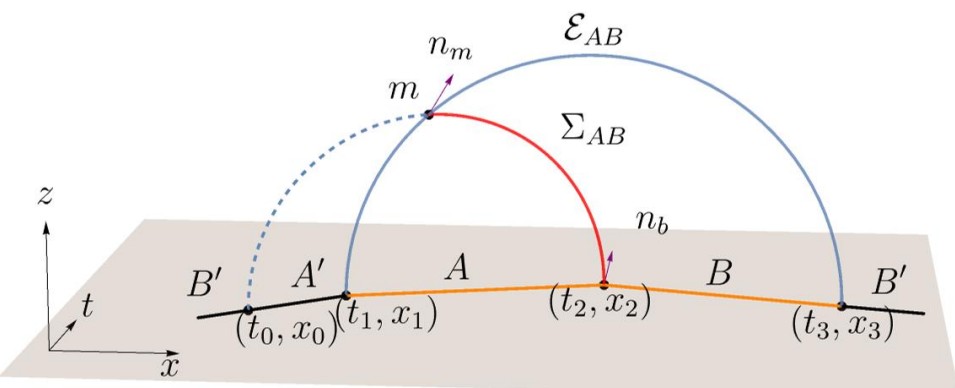

Figure 5: Illustration of EWCS in the adjacent case. The red curve is the EWCS $\Sigma_{AB}$ whose extension (the dashed blue line) ends at the balance point $(t_0, x_0)$. The vector $n_m$ is normal to both the EWCS and $\mathcal{E}_{AB}$, while $n_b \propto \partial_t$.

### 4.2.2 Adjacent cases

For our purpose, the setup for adjacent $AB$ is illustrated in Fig.5. A point on the RT curve $\mathcal{E}_{AB}$ from $(U_1, V_1, \infty)$ to $(U_3, V_3, \infty)$ can be parametrized by:

$$U_k = \frac{U_3 - U_1}{2}k + \frac{U_3 + U_1}{2}, \quad V_k = \frac{V_3 - V_1}{2}k + \frac{V_3 + V_1}{2},$$
$$\rho_k = \frac{2}{(U_3 - U_1)(V_3 - V_1)(1 - k^2)}, \tag{78}$$

where $k = \tanh \tau$ such that $(U_1, V_1, \infty)$ corresponds to $k = -1$ and $(U_3, V_3, \infty)$ corresponds to $k = 1$. According to our previous discussions, $E_W^{\text{EH}}$ is given by the length of geodesic segment $\Sigma_{AB}$ between $(U_2, V_2, \rho_2 \to \infty)$ and the point $(U_m, V_m, \rho_m)$ which we denote as $m$:

$$E_W^{\text{EH}} = \frac{L_{\text{AdS}}(U_m, V_m, \rho_m, U_2, V_2, \rho_2)}{4G}, \tag{79}$$

where $(U_m, V_m, \rho_m)$ satisfies

$$\frac{\partial}{\partial k} L_{\text{AdS}}(U_k, V_k, \rho_k, U_2, V_2, \rho_2)\big|_{k=m} = 0, \tag{80}$$

and $k = m$ symbolically means that $(U_k, V_k, \rho_k) = (U_m, V_m, \rho_m)$. The above condition demonstrates that the segment $\Sigma_{AB}$ is extremal. We further set $(U_2, V_2) = (0, 0)$ without losing generality, the solution of the above saddle condition is given by

$$k_{saddle} = \frac{U_1 V_1 \rho_2 - U_3 V_3 \rho_2}{1 + U_1 V_1 \rho_2 + U_3 V_3 \rho_2} \to \frac{U_1 V_1 - U_3 V_3}{U_1 V_1 + U_3 V_3}, \tag{81}$$

in the limit of $\rho_2 \to \infty$. We therefore have

$$U_m = \frac{U_1 U_3 (V_1 + V_3)}{U_1 V_1 + U_3 V_3},$$
$$V_m = \frac{V_1 V_3 (U_1 + U_3)}{U_1 V_1 + U_3 V_3}, \tag{82}$$
$$\rho_m = \frac{(U_1 V_1 + U_3 V_3)^2}{2U_1 V_1 U_3 V_3 (U_1 - U_3)(V_1 - V_3)}.$$

Plugging the point $m$ into the length formula (75) for geodesic chords, we have

$$E_W^{\text{EH}} = \frac{1}{8G} \log\left(\frac{8U_1 U_3 V_1 V_3 \rho_2}{(U_3 - U_1)(V_3 - V_1)}\right). \tag{83}$$

In order to match with the BPE (27), we should transfer to the parameters we used in the last section by using (71) and the following relations,

$$x_1 = -R_A \cosh\kappa_A, \quad t_1 = -R_A \sinh\kappa_A, \quad x_3 = R_B \cosh\kappa_A, \quad t_3 = R_B \sinh\kappa_B, \tag{84}$$

$$\tanh\kappa_{AB} = \frac{R_A \sinh\kappa_A + R_B \sinh\kappa_B}{R_A \cosh\kappa_A + R_B \cosh\kappa_B},$$
$$R_{AB}^2 = R_A^2 + R_B^2 + 2R_A R_B \cosh(\kappa_A - \kappa_B). \tag{85}$$

At last we arrive at

$$E_W^{\text{EH}} = \frac{1}{4G} \log\left(\frac{2R_A R_B}{\delta R_{AB}}\right) = \frac{c_{geo}}{12} \log\left(\frac{2R_A R_B}{\delta R_{AB}}\right), \tag{86}$$

where $\delta$ comes from the cut-off $z_2 = \delta$. We can clearly see that $E_W^{\text{EH}}$ coincides with the *geometric* term of (27).

The EWCS has a direct relation with balance conditions in a way that the EWCS is part of the spacelike geodesic connecting the solutions of balance conditions, i.e. the extension of $\Sigma_{AB}$ reaches the balance point as illustrated in Fig.5. To be more concrete, we rewrite (26) in the light-cone coordinate:

$$U_0 = \frac{2U_1 U_3}{U_1 + U_3}, \quad V_0 = \frac{2V_1 V_3}{V_1 + V_3}, \tag{87}$$

such that the geodesic from $(U_2, V_2, \rho_2) = (0, 0, \infty)$ to $(U_0, V_0, \infty)$ is given by [7]

$$U_s = \frac{U_1 U_3}{U_1 + U_3}s + \frac{U_1 U_3}{U_1 + U_3}, \quad V_s = \frac{V_1 V_3}{V_1 + V_3}s + \frac{V_1 V_3}{V_1 + V_3},$$
$$\rho_s = \frac{(U_1 + U_3)(V_1 + V_3)}{2(1 - s^2)U_1 U_3 V_1 V_3}. \tag{88}$$

On this geodesic, one can verify that when

$$s = \frac{U_3 V_1 + U_1 V_3}{U_1 V_1 + U_3 V_3}, \tag{89}$$

the corresponding point is exactly the saddle point $m$ given by (82).

### 4.2.3 Non-adjacent cases

We consider a non-adjacent configuration illustrated in Fig.6. When the entanglement wedge is connected, there are two RT curves $\mathcal{E}_1, \mathcal{E}_2$, with $\mathcal{E}_1$ connecting $(U_2, V_2, \infty)$ and $(U_4, V_4, \infty)$ parametrized by

$$U_g = \frac{U_4 - U_2}{2}g + \frac{U_4 + U_2}{2}, \quad V_g = \frac{V_4 - V_2}{2}g + \frac{V_4 + V_2}{2},$$
$$\rho_g = \frac{2}{(U_4 - U_2)(V_4 - V_2)(1 - g^2)}, \tag{90}$$

---

[7]In this section there are two different geodesics (parametrized by $k$ and $s$ respectively) from which one should distinguish.

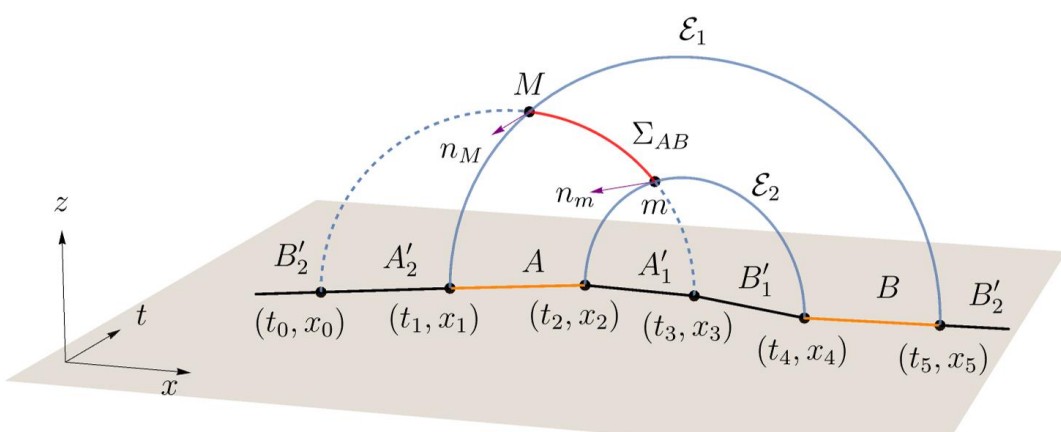

Figure 6: Illustration of EWCS in the non-adjacent case. The red curve is the EWCS $\Sigma_{AB}$ whose extension (the dashed blue line) ends at the balance points $(t_0, x_0)$ and $(t_3, x_3)$ and the RT surface is disconnected $\mathcal{E}_{AB} = \mathcal{E}_1 \cup \mathcal{E}_2$. The vectors $n_{M,m}$ are normal to both the EWCS and $\mathcal{E}_{1,2}$.

and $\mathcal{E}_2$ connecting $(U_1, V_1, \infty)$ and $(U_5, V_5, \infty)$ parametrized by

$$U_h = \frac{U_5 - U_1}{2}h + \frac{U_5 + U_1}{2}, \quad V_h = \frac{V_5 - V_1}{2}h + \frac{V_5 + V_1}{2},$$
$$\rho_h = \frac{2}{(U_5 - U_1)(V_5 - V_1)(1 - h^2)}. \tag{91}$$

The lengths of all the geodesic chords connecting the two RT curves can be calculated by substituting the above parametrizations into (75). To calculate $E_W^{\mathrm{EH}}$, we need to impose extremal conditions with respect to $g, h$ respectively:

$$\frac{\partial}{\partial g} L_{\mathrm{AdS}}\left(U_g, V_g, \rho_g, U_h, V_h, \rho_h\right) = 0, \tag{92}$$

$$\frac{\partial}{\partial h} L_{\mathrm{AdS}}\left(U_g, V_g, \rho_g, U_h, V_h, \rho_h\right) = 0. \tag{93}$$

The solutions $g = g_{saddle}$, $h = h_{saddle}$ to the above saddle equations gives the two endpoints $m$ and $M$ of the EWCS respectively, whose length can be calculated by (75),

$$E_W^{\mathrm{EH}} = \frac{L_{\mathrm{AdS}}(U_m, V_m, \rho_m, U_M, V_M, \rho_M)}{4G}. \tag{94}$$

Specifically we consider the setup of **Case 1** (51) with $(U_1, V_1) = (0, 0)$ and set $(U_5, V_5)$ or $\eta$, $\bar{\eta}$ free. It is useful to introduce parameters $\zeta, \bar{\zeta}$:

$$\zeta \equiv \frac{\sqrt{\eta} + 1}{\sqrt{\eta} - 1}, \quad \bar{\zeta} \equiv \frac{\sqrt{\bar{\eta}} + 1}{\sqrt{\bar{\eta}} - 1}. \tag{95}$$

We find the solutions are given by

$$g_{saddle} = -\frac{4 + 5\zeta + 5\bar{\zeta} + 4\zeta\bar{\zeta}}{5 + 4\zeta + 4\bar{\zeta} + 5\zeta\bar{\zeta}}, \tag{96}$$

$$s_{saddle} = \frac{\bar{\zeta}(2 + \bar{\zeta}) + \zeta^2(1 + 2\bar{\zeta}) + 2\zeta(1 + \bar{\zeta}(4 + \bar{\zeta}))}{1 + 2\bar{\zeta} + \zeta(2 + 2(4 + \zeta)\bar{\zeta} + (2 + \zeta)\bar{\zeta}^2)}. \tag{97}$$

Then, in terms of $\eta, \bar{\eta}$ the two endpoints of the EWCS are given by:

$$U_m = \frac{1}{2} + \frac{1}{1 + 9\sqrt{\eta\bar{\eta}}} \, ,$$

$$V_m = \frac{1}{2} + \frac{1}{1 + 9\sqrt{\eta\bar{\eta}}} \, , \tag{98}$$

$$\rho_m = \frac{(1 + 9\sqrt{\eta\bar{\eta}})^2}{18\sqrt{\eta\bar{\eta}}} \, ,$$

and

$$U_M = \frac{3(-1 + 3\eta)(-1 + \bar{\eta})}{2 + 8\sqrt{\eta\bar{\eta}} - 6\bar{\eta} + 6\eta(-1 + 3\bar{\eta})} \, ,$$

$$V_M = \frac{3(-1 + \eta)(-1 + 3\bar{\eta})}{2 + 8\sqrt{\eta\bar{\eta}} - 6\bar{\eta} + 6\eta(-1 + 3\bar{\eta})} \, , \tag{99}$$

$$\rho_M = \frac{(1 + 4\sqrt{\eta\bar{\eta}} - 3\bar{\eta} + \eta(-3 + 9\bar{\eta}))^2}{18\sqrt{\eta\bar{\eta}}(-1 + \eta)(-1 + \bar{\eta})} \, .$$

Eventually we arrive at

$$E_W^{\text{EH}} = \frac{1}{8G} \log(\zeta\bar{\zeta}) = \frac{c_{geo}}{24} \log \frac{(\sqrt{\eta} + 1)(\sqrt{\bar{\eta}} + 1)}{(\sqrt{\eta} - 1)(\sqrt{\bar{\eta}} - 1)} \, , \tag{100}$$

which coincides with the *geometric* term of (54). Other cases can be verified in a similar manner. One can arrive at the same formula starting from other setup different from **Case 1**.

Similar to the adjacent case, the EWCS in the non-adjacent case is part of the geodesic connecting the two partition points $Q_{1,2}$ (see Fig.6), which are previously determined by balance conditions with the help of gravitational anomalies. We rewrite (50) and (52):

$$U_0 = \frac{1}{2} + \frac{1}{1 - 3\sqrt{\bar{\eta}}} \, , \quad V_0 = \frac{1}{2} + \frac{1}{1 - 3\sqrt{\eta}} \, , \tag{101}$$

$$U_3 = \frac{1}{2} + \frac{1}{1 + 3\sqrt{\bar{\eta}}} \, , \quad V_3 = \frac{1}{2} + \frac{1}{1 + 3\sqrt{\eta}} \, . \tag{102}$$

The geodesic parameterized by $\omega$ from $(U_0, V_0, \infty)$ to $(U_3, V_3, \infty)$ is given by:

$$U_\omega = \frac{3\sqrt{\bar{\eta}}}{-1 + 9\bar{\eta}} \omega + \frac{1}{2} + \frac{1}{1 - 9\bar{\eta}} \, ,$$

$$V_\omega = \frac{3\sqrt{\eta}}{-1 + 9\eta} \omega + \frac{1}{2} + \frac{1}{1 - 9\eta} \, , \tag{103}$$

$$\rho_\omega = \frac{(-1 + 9\eta)(-1 + 9\bar{\eta})}{18(1 - \omega^2)\sqrt{\eta\bar{\eta}}} \, .$$

One can check that the two end points of the EWCS $M$ (99) and $m$ (98) lie exactly on this geodesic with

$$\omega = \omega_M \equiv \frac{2(\sqrt{\eta}(1 - 3\bar{\eta}) + \sqrt{\bar{\eta}} - 3\eta\sqrt{\bar{\eta}})}{1 + 4\sqrt{\eta\bar{\eta}} - 3\bar{\eta} + \eta(-3 + 9\bar{\eta})} \, , \tag{104}$$

$$\omega = \omega_m \equiv \frac{3(\sqrt{\eta} + \sqrt{\bar{\eta}})}{1 + 9\sqrt{\eta\bar{\eta}}} \, . \tag{105}$$

This again confirms that, the partition points for $A'B'$ satisfying the balance requirements are exactly where the extension of the EWCS anchors on the boundary.

## 4.3 Correction to the EWCS from gravitational anomalies

### 4.3.1 Prescription

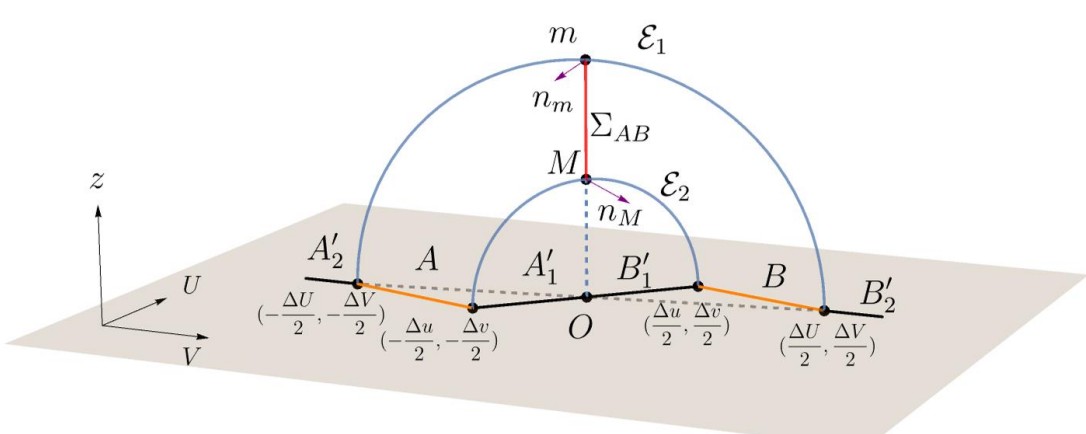

Figure 7: Illustration of EWCS for the two non-adjacent symmetric intervals.

Unlike the RT surfaces, the EWCS is a geodesic chord in the bulk with the endpoints not anchored on the asymptotic boundary. In such scenarios, the boundary conditions are not useful to determine how much the normal vector $n$ is twisted along the geodesic between the initial and the final endpoints of the EWCS. In other words, the second integration in (68) on the EWCS is coordinate dependent, which can no longer be fixed by the boundary conditions. This makes the proposal (68) un-physical and brings new challenge to calculate or even define the contribution from the CS term to the entropy quantity $E_W(A, B)$.

In this subsection, we will first show that the dependence of the coordinates for the integration on the EWCS is indeed needed for us to reproduce the BPE or the reflected entropy in the dual field theory. Then we will give a new geometric prescription to calculate the correction to the EWCS from the CS term. The prescription will tell us how to properly choose the $n$ vectors at the endpoints of the EWCS according to the intervals $A$ and $B$.

Previously we show that the BPE and the reflected entropy consist of two parts, the *geometric* term and the *anomalous* term. Let us consider the symmetric configuration illustrated in Fig.7 where the endpoints of $A$ and $B$ have reflection symmetry with respect to the origin $O$. When the entanglement wedge is connected, the EWCS, which is defined as the saddle geodesic anchored on the two disconnected RT curves $\mathcal{E}_{1,2}$, lies along the $\rho$ coordinate and anchors on the two turning points of $\mathcal{E}_{1,2}$. It is interesting that, when we change $\Delta U$ and $\Delta V$ while keeping the configuration symmetric under reflection with respect to the origin $O$, the RT surface $\mathcal{E}_1$ will rotate with the turning point $m$ fixed. Also the turning point $M$ of $\mathcal{E}_2$ is fixed when we rotate the interval $A'_1 \cup B'_1$ by adjusting $\Delta u$ and $\Delta v$.

An important observation we get is that, under the rotations we mentioned above, the segment $\Sigma_{AB}$, as well as $E_W^{EH}(A, B)$, are fixed, while the *anomalous* term of the BPE changes. To be more specific, we have $\kappa_A = \kappa_B$. If we only adjust $\Delta u$ and $\Delta v$, then $\kappa_{AB}$ is fixed while $\kappa_A$ changes. This definitely changes the value of $(\kappa_A + \kappa_B - \kappa_{AB})$ which is proportional to the *anomalous* term of the BPE. In summary, one geodesic chord in the bulk could correspond to the BPE (or the reflected entropy) of different mixed states, which have different values. This contrasts with our experience for the cases without gravitational anomalies that the entropy quantity associated to the EWCS is totally determined by the gravity theory and the geometry.

Thanks to the coordinate dependence of the integral $\int_{\Sigma_{AB}} d\tau\, \tilde{n}\cdot\nabla n$, there is room to adjust the normal vectors $n$ at the endpoints of the EWCS. A further input that determines the $n$ and $\tilde{n}$ vectors at the endpoints should come from the mixed state. Note that, the endpoints of the EWCS are anchored on the RT surface of the mixed state system $AB$, which may contain the information we need to settle down the normal frame. We claim that, the three normalized vectors $v$ (spacelike), $n$ (timelike) and $\tilde{n}$ (spacelike) that determine the normal frame at the endpoints of the EWCS are chosen by the following *prescription*:

- $v$ is the vector tangent to the EWCS;

- $\tilde{n}$ is the vector tangent to the RT surface where the endpoint of the EWCS is anchored;

- $n$ is determined by the above $v$ and $\tilde{n}$, which is a normalized vector normal to both of the EWCS and the RT surface of $AB$.

Both $n, \tilde{n}$ can be determined up to an overall sign related a choice of handedness. Here we only need choices of the signs to ensure that the term inside logarithm of (66) is positive. We will give further discussions on this point in the discussion section. With the normal vectors properly chosen, the correction to the EWCS from the CS term is then straightforwardly given by (66). For example, the two vectors $n_m$ and $n_M$ in Fig.7 are chosen following our prescription. In the following we will apply the above prescription to generic configurations of $AB$ and show that, the integration $E_W^{CS}(A,B)$ will reproduce the *anomalous* term of the BPE.

### 4.3.2 Adjacent cases

Let us consider again the configuration illustrated in Fig.5 and set $(U_2, V_2) = (0, 0)$ without loss of generality, one endpoint of $\Sigma_{AB}$ intersects with $\mathcal{E}_{AB}$ while the other one anchors on the boundary. By using (74) for two geodesics (78) (88) at the interaction point $m$, two tangent vectors are given by:

$$
\begin{aligned}
v_{\Sigma_{AB}} =& \frac{U_1 U_3 (U_1 - U_3)(V_1^2 - V_3^2)}{(U_1 V_1 + U_3 V_3)^2} \partial_U + \frac{V_1 V_3 (U_1^2 - U_3^2)(V_1 - V_3)}{(U_1 V_1 + U_3 V_3)^2} \partial_V \\
&+ \frac{(U_3 V_1 + U_1 V_3)(U_1 V_1 + U_3 V_3)}{U_1 U_3 V_1 V_3 (U_1 - U_3)(V_1 - V_3)} \partial_\rho \,, \\
v_{\mathcal{E}_{AB}} =& \frac{2 U_1 U_3 V_1 V_3 (-U_1 + U_3)}{(U_1 V_1 + U_3 V_3)^2} \partial_U + \frac{2 U_1 U_3 V_1 V_3 (-V_1 + V_3)}{(U_1 V_1 + U_3 V_3)^2} \partial_V \\
&+ \frac{(U_1 V_1 - U_3 V_3)(U_1 V_1 + U_3 V_3)}{U_1 U_3 V_1 V_3 (U_1 - U_3)(V_1 - V_3)} \partial_\rho \,,
\end{aligned}
\tag{106}
$$

when $k$ takes the value of (81) while $s$ takes the value of (89) for the corresponding tangent vectors respectively. We can directly verify that they are normal to each other. According to our prescription,

$$
\tilde{n}_m = v_{\mathcal{E}_{AB}} \,,
\tag{107}
$$

such that $n$ can be obtained after solving

$$
n_m \cdot \tilde{n}_m = 0 \,, \quad n_m \cdot v_{\Sigma_{AB}} = 0 \,, \quad n_m^2 = -1 \,.
\tag{108}
$$

The solution is given by (up to a minus sign in the first two components):

$$
\begin{aligned}
n_m =& \pm \frac{\sqrt{F}(V_1^2 + V_3^2)}{V_1 (V_1 - V_3) V_3 H^2} \partial_U \mp \frac{\sqrt{F}(U_1^2 + U_3^2)}{U_1 (U_1 - U_3) U_3 H^2} \partial_V \\
&+ \frac{(U_3 V_1 - U_1 V_3) H}{\sqrt{F}} \partial_\rho \,, \\
F \equiv& U_1^2 U_3^2 V_1^2 V_3^2 (U_1 - U_3)^2 (V_1 - V_3)^2 \,, \\
H \equiv& U_1 V_1 + U_3 V_3 \,.
\end{aligned}
\tag{109}
$$

For the endpoint $(U_2, V_2, \infty)$ on the boundary, $n$ is the future-directing timelike vector for the dual CFT as discussed above:

$$n_b = z\partial_t = \frac{1}{\sqrt{2\rho}}\left(\partial_U - \partial_V\right). \tag{110}$$

Indeed this choice can not be explained following our prescription since the RT surface $\mathcal{E}_{AB}$ vanishes here. It only corresponds to certain limit for the non-adjacent cases. We will give further discussion at this point in the last section. The right choice of $n_m$ in (109) should make the term inside logarithm of $E_W^{CS}$ positive, and we find that the one with plus sign in the first component does satisfy the requirement. Using (71)(84)(85) and applying (77) to (88) to calculate the reference normal frame $q, \tilde{q}$ along $\Sigma_{AB}$, we have[8]

$$\frac{q_m \cdot n_m - \tilde{q}_m \cdot n_m}{q_b \cdot n_b - \tilde{q}_b \cdot n_b} = \frac{e^{\kappa_B}R_A + e^{\kappa_A}R_B}{R_{AB}}, \tag{111}$$

which gives

$$\log\left(\frac{q_m \cdot n_m - \tilde{q}_m \cdot n_m}{q_b \cdot n_b - \tilde{q}_b \cdot n_b}\right) = \frac{1}{2}\left(\kappa_A + \kappa_B + \log\frac{e^{\kappa_B}R_A + e^{\kappa_A}R_B}{e^{\kappa_A}R_A + e^{\kappa_B}R_B}\right) \tag{112}$$
$$= \kappa_A + \kappa_B - \kappa_{AB},$$

we therefore have:

$$E_W^{CS} = \frac{1}{4G\mu}\left(\kappa_A + \kappa_B - \kappa_{AB}\right) = -\frac{c_{ano}}{12}\left(\kappa_A + \kappa_B - \kappa_{AB}\right), \tag{113}$$

which coincides with the *anomalous* term of (27).

### 4.3.3 Non-adjacent cases

Again we consider the configuration illustrated in Fig.6 and focus on **Case 1** (51) with $(U_1, V_1) = (0, 0)$, the two RT curves $\mathcal{E}_1, \mathcal{E}_2$ under consideration are parametrized by (90) (91), and the endpoints $m$, $M$ of $\Sigma_{AB}$ are given by (98) and (99) respectively. Each endpoint of $\Sigma_{AB}$ intersects with one RT curve and we can apply (74) for both the RT curves such that our prescription gives

$$\tilde{n}_m = v_{\mathcal{E}_2} = \frac{18\sqrt{\eta}\sqrt{\bar{\eta}}}{(1 + 9\sqrt{\eta}\sqrt{\bar{\eta}})^2}\partial_U + \frac{18\sqrt{\eta}\sqrt{\bar{\eta}}}{(1 + 9\sqrt{\eta}\sqrt{\bar{\eta}})^2}\partial_V + \frac{1 - 81\eta\bar{\eta}}{9\sqrt{\eta}\sqrt{\bar{\eta}}}\partial_\rho,$$

$$\tilde{n}_M = v_{\mathcal{E}_1} = \frac{12\sqrt{\eta}(-1 + 3\eta)(-1 + \bar{\eta})\sqrt{\bar{\eta}}}{(1 + 4\sqrt{\eta}\sqrt{\bar{\eta}} - 3\bar{\eta} + \eta(-3 + 9\bar{\eta}))^2}\partial_U + \frac{12(-1 + \eta)\sqrt{\eta}\sqrt{\bar{\eta}}(-1 + 3\bar{\eta})}{(1 + 4\sqrt{\eta}\sqrt{\bar{\eta}} - 3\bar{\eta} + \eta(-3 + 9\bar{\eta}))^2}\partial_V$$

$$+ \frac{(1 - 3\bar{\eta})^2 + 9\eta^2(1 - 3\bar{\eta})^2 + \eta\left(-6 + 20\bar{\eta} - 54\bar{\eta}^2\right)}{9(-1 + \eta)\sqrt{\eta}(-1 + \bar{\eta})\sqrt{\bar{\eta}}}\partial_\rho, \tag{114}$$

at the endpoints $m$ and $M$ respectively. Next the $n$ vector at $m$ and $M$ can be found by solving

$$n_m \cdot \tilde{n}_m = 0, \quad n_m \cdot v_{\Sigma_{AB}}\big|_m = 0, \quad n_m^2 = -1, \tag{115}$$
$$n_M \cdot \tilde{n}_M = 0, \quad n_M \cdot v_{\Sigma_{AB}}\big|_M = 0, \quad n_M^2 = -1,$$

---

[8]Note that with parametrization (88), the initial endpoint is $(U_2, V_2, \infty)$ and the final endpoint is $(U_m, V_m, \rho_m)$.

where $v_{\Sigma_{AB}}\big|_m$, $v_{\Sigma_{AB}}\big|_M$ are tangent vectors of $\Sigma_{AB}$ (103) at $m$ and $M$ respectively, given by

$$
\begin{aligned}
v_{\Sigma_{AB}}\big|_m &= \frac{3(-1+9\eta)\sqrt{\bar{\eta}}}{(1+9\sqrt{\eta}\sqrt{\bar{\eta}})^2}\partial_U + \frac{3\sqrt{\eta}(-1+9\bar{\eta})}{(1+9\sqrt{\eta}\sqrt{\bar{\eta}})^2}\partial_V + \frac{1}{3}\left(\frac{1+9\eta}{\sqrt{\eta}}+\frac{1+9\bar{\eta}}{\sqrt{\bar{\eta}}}\right)\partial_\rho\,, \\
v_{\Sigma_{AB}}\big|_M &= \frac{3(-1+\eta)(-1+9\eta)(-1+\bar{\eta})\sqrt{\bar{\eta}}}{(1+4\sqrt{\eta}\sqrt{\bar{\eta}}-3\bar{\eta}+\eta(-3+9\bar{\eta}))^2}\partial_U + \frac{3(-1+\eta)\sqrt{\eta}(-1+\bar{\eta})(-1+9\bar{\eta})}{(1+4\sqrt{\eta}\sqrt{\bar{\eta}}-3\bar{\eta}+\eta(-3+9\bar{\eta}))^2}\partial_V \\
&\quad -\frac{2(\sqrt{\eta}+\sqrt{\bar{\eta}})(-1+3\sqrt{\eta}\sqrt{\bar{\eta}})(1+4\sqrt{\eta}\sqrt{\bar{\eta}}-3\bar{\eta}+\eta(-3+9\bar{\eta}))}{9(-1+\eta)\sqrt{\eta}(-1+\bar{\eta})\sqrt{\bar{\eta}}}\partial_\rho\,.
\end{aligned}
\tag{116}
$$

We therefore have:

$$
n_m = \mp\frac{3(1+9\eta)\sqrt{\bar{\eta}}}{(1+9\sqrt{\eta\bar{\eta}})^2}\partial_U \pm \frac{3\sqrt{\eta}(1+9\bar{\eta})}{(1+9\sqrt{\eta\bar{\eta}})^2}\partial_V + \frac{1}{3}\left(\frac{1-9\eta}{\sqrt{\eta}}+\frac{-1+9\bar{\eta}}{\sqrt{\bar{\eta}}}\right)\partial_\rho\,,
\tag{117}
$$

$$
\begin{aligned}
n_M &= \pm\frac{3(\eta(9\eta-2)+1)(\bar{\eta}-1)\sqrt{\bar{\eta}}}{(\eta(9\bar{\eta}-3)+4\sqrt{\eta}\sqrt{\bar{\eta}}-3\bar{\eta}+1)^2}\partial_U \mp \frac{3(\eta-1)\sqrt{\eta}(\bar{\eta}(9\bar{\eta}-2)+1)}{(\eta(9\bar{\eta}-3)+4\sqrt{\eta}\sqrt{\bar{\eta}}-3\bar{\eta}+1)^2}\partial_V \\
&\quad -\frac{2(\sqrt{\eta}-\sqrt{\bar{\eta}})(3\sqrt{\eta}\sqrt{\bar{\eta}}+1)(\eta(9\bar{\eta}-3)+4\sqrt{\eta}\sqrt{\bar{\eta}}-3\bar{\eta}+1)}{9(\eta-1)\sqrt{\eta}(\bar{\eta}-1)\sqrt{\bar{\eta}}}\partial_\rho\,.
\end{aligned}
\tag{118}
$$

Choosing the up sign of the above solutions and applying (77) to (103) we find

$$
\log\left(\frac{q_m\cdot n_m-\tilde{q}_m\cdot n_m}{q_M\cdot n_M-\tilde{q}_M\cdot n_M}\right) = -\frac{1}{2}\log\frac{\left(\sqrt{\eta}+1\right)\left(\sqrt{\bar{\eta}}-1\right)}{\left(\sqrt{\eta}-1\right)\left(\sqrt{\bar{\eta}}+1\right)}\,,
\tag{119}
$$

such that

$$
E_W^{\text{CS}} = -\frac{1}{8G\mu}\log\frac{\left(\sqrt{\eta}+1\right)\left(\sqrt{\bar{\eta}}-1\right)}{\left(\sqrt{\eta}-1\right)\left(\sqrt{\bar{\eta}}+1\right)} = \frac{c_{ano}}{24}\log\frac{\left(\sqrt{\eta}+1\right)\left(\sqrt{\bar{\eta}}-1\right)}{\left(\sqrt{\eta}-1\right)\left(\sqrt{\bar{\eta}}+1\right)}\,,
\tag{120}
$$

which is exactly the *anomalous* term of (54). One can check that, for other configurations, like **Case 2,3** (55), the anomalous term will be given by the same formula.

# 5  Summary and discussion

## 5.1  Summary for the main results

- We calculated the BPE in the $\text{CFT}_2$ with and without gravitational anomalies, and find they coincide exactly with the reflected entropies. This is a non-trivial test to the claim that the BPE is purification independent and captures the same type of mixed state correlations as the reflected entropy.

- Using the correspondence between the $\text{CFT}_2$ with gravitational anomalies and locally AdS geometries in TMG, we explore the holographic picture for the BPE and the reflected entropy, which is the entropy quantity associated to the EWCS. We extend the concept of the EWCS to be the saddle geodesic chord connecting the different pieces of the disconnected RT surface $\mathcal{E}_{AB}$. We calculated the length of the covariant EWCS and find it reproduce the part of the BPE proportional to $c_L + c_R$.

- We expect the correction to the EWCS from the CS term in TMG should reproduce the part of the BPE and the reflected entropy originated from the gravitational anomalies,

i.e. the correction is proportional to $c_L - c_R$. Based on this expectation, we find that the gravity theory and the near curve geometry is not enough to capture the anomalous part of the BPE. Further input from the mixed state $\rho_{AB}$ should be taken into account, which is just carried by the RT surface $\mathcal{E}_{AB}$. Our prescription requires the vector $n$ at the endpoints of the EWCS to be normal to both the $\mathcal{E}_{AB}$ and the EWCS.

Our results give further clear evidence to the conjecture that, the BPE is an intrinsic measure for mixed state correlations and duals to the EWCS in holography. The BPE exactly matches with the reflected entropy in covariant scenarios with and without gravitational anomalies. The minimized crossing PEE (or the Markov gap [71] in the case of canonical purification ), which is conjectured to be universal for the adjacent cases, receives no contribution from the gravitational anomaly. Although, the partition points of the purifying system are determined with the help of the anomalies, the position of the partition points does not depend on the anomalies. On the other hand, in holographic theories with geometric description, the partition points can also be determined by extending the EWCS, with no reference to the gravitational anomalies. We may conclude that the existence of the gravitational anomalies is not essential for us to calculate the BPE for the covariant configurations.

More importantly, we gave a novel prescription to evaluate the corrections to the EWCS from the CS term in TMG. This is indeed the first explicit calculation on the corrections to the EWCS in theories beyond Einstein gravity[9]. Despite little was explored, this could be an important research direction in the future. The non-covariance property of TMG makes this evaluation even more challenging, and makes the prescription not intrinsic on the gravity side. Although this is surprising to us, it may not happen in other higher derivative gravities which have covariance. Nevertheless, we see that this non-intrinsic property of our prescription is indeed necessary to reproduce the results for the BPE and the reflected entropy with gravitational anomalies.

Currently the entanglement negativity attracts considerable attention from both of the condensed matter community and the high energy community. However, the way the negativity is defined is quite different from the reflected entropy and BPE, hence it is difficult to see how these quantities are related and differ from each other. In AdS/CFT the three quantities are all claimed to be dual to the EWCS. Recently, the negativity in $CFT_2$ with gravitational anomaly is also carried out in [44] using the monodromy techniques. However the results do not match with the EWCS. For example, in the adjacent case the negativity differ from the EWCS by a constant of order $c$, which is just the balanced crossing PEE. On the other hand the negativity calculated by the correlation functions of twist operators [13] gives a more accurate matching with the EWCS and the reflected entropy. All in all, as was pointed out in [85], our current understanding of the entanglement negativity have not led to a clear geometric picture yet. We hope to revisit to this point and get further understanding in the future.

## 5.2 More on the prescription

For the covariant configurations without gravitational anomalies, the correspondence between the BPE and the EWCS can be demonstrated by the fine correspondence [28] between the points on the boundary interval and the points on the RT surface. See the arguments in section 6 of [1] for the static configurations. The fine correspondence is originated from slicing the entanglement wedge with the so-called modular slices [28], and the claim is that the contribution from any site inside $A$ to $S_A$ is represented by its partner point on the RT surface. For

---

[9]In [44] the authors also considered the EWCS in TMG in the symmetric configurations shown by Fig.7. They used the $n$ vector given in [84] which have an explicit formula along the whole EWCS, and did the integration $\int_{\Sigma_{AB}} d\tau \, \tilde{n} \cdot \nabla n$ along the EWCS. Nevertheless, we found their calculation not consistent and eventually will not reproduce the reflected entropy in general cases. See the erratum of [44].

each point on $A$, if we consider the geodesic emanating from it and intersects with $\mathcal{E}_{AB}$ vertically, the intersection point on $\mathcal{E}_{AB}$ is the partner point. This partnership was only explicitly discussed in the static cases, but we see no obstacle to extend this partnership to the covariant cases as long as the modualr flow is local. The fine correspondence further indicates that for any subinterval $A_i$ inside $A$, the PEE $s_A(A_i)$ is given by the length of a geodesics chord $\mathcal{E}_i$, which consists of all the partner points of the points in $A_i$, i.e, a correspondence between geodesics chords and PEE [28].

$$s_A(A_i) = \frac{Length(\mathcal{E}_i)}{4G}. \tag{121}$$

For example, consider the configuration in Fig.8, where the extension of $\Sigma_{AB}$ determines the partition of the purifying system $A'B'$. The extended $\Sigma_{AB}$ gives the entanglement entropy for the interval $A'_1AA'_2$. Since $\mathcal{E}_{AB}$ and $\Sigma_{AB}$ are normal to each other, according to the fine correspondence, the partners for the points inside $A$ are just those settled on $\Sigma_{AB}$,

$$s_{A'_1AA'_2}(A) = \frac{Length(\Sigma_{AB})}{4G}. \tag{122}$$

On the other hand, the extended $\Sigma_{AB}$ is also the RT surface of $B'_1BB'_2$. According to the fine correspondence in $\mathcal{W}_{BB'}$ we also have,

$$s_{B'_1BB'_2}(B) = \frac{Length(\Sigma_{AB})}{4G}. \tag{123}$$

The observation that the same EWCS $\Sigma_{AB}$ corresponds to two PEEs, $s_{AA'}(A)$ and $s_{BB'}(B)$, is a manifestation of the balance condition. Together with (122) and (123), we arrive at the correspondence between $BPE(A,B)$ and the EWCS.

Next we include the gravitational anomalies and ask whether the balance condition can help us choose the right $n$ previously given in our prescription. As we have shown that, the same EWCS $\Sigma_{AB}$ corresponds to two different PEEs $s_{AA'}(A)$ and $s_{BB'}(B)$, depending on whether we apply the fine correspondence in the entanglement wedge of $AA'$ or $BB'$. With the entanglement wedge chosen, we need to set up a standard for choosing the direction (sign) of the vectors in the normal frame. Here we choose the direction of $v$ by requiring that when walking along $v$, the entanglement wedge should lie on the left-hand side of $v$. At this point we only require that $\tilde{n}$ points inwards the entanglement wedge. Then $n$ is determined by

$$n = v \times \tilde{n}. \tag{124}$$

What we want to emphasize is that, the normal frames chosen by the same standard from different sides of $\mathcal{E}_{AB}$ should show a reflection symmetry with respect to $\mathcal{E}_{AB}$ in the near curve region and give the same $n$ from each side. This reflection symmetry of the normal frames is again a manifestation of the balance requirement for the anomalous contribution, which gives us a hint on how to determine $\tilde{n}$, and also $n$. Although, the symmetry is indeed not manifest globally in configurations like Fig.8, but will appear after a conformal transformation which lead to the Rindler bulk, where the $AA'$ and $BB'$ becomes the two sides of an eternal black hole [86]. In the canonical purifications discussed in [9], the reflection symmetry is obvious. With this local reflection symmetry, the choice that $\tilde{n}$ should be tangent to $\mathcal{E}_{AB}$ is the only one that leads to the same vector $n$ from both sides. Following this standard, if we consider the entanglement wedge $\mathcal{W}_{AA'}$, the vectors in the normal frame are shown by the black arrows in Fig.8. If we consider $\mathcal{W}_{BB'}$ from the other side, the $v$ and $\tilde{n}$ vectors will change their sign, while $n$ remains unchanged, as well as the integral (66).

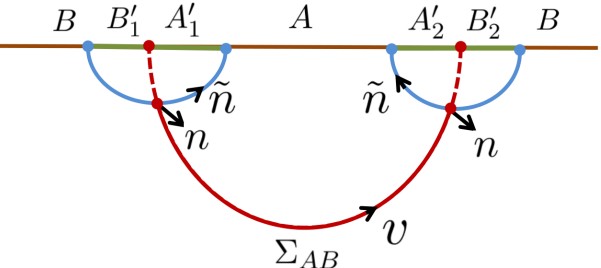

Figure 8: The blue lines are the RT surfaces $\mathcal{E}_{AB}$ while the red curve is the EWCS $\Sigma_{AB}$ which is normal to $\mathcal{E}_{AB}$. Though the figure looks static, we should consider it to be a covariant configuration.

## 5.3 Reproducing the entanglement entropy from the EWCS

The last topic we would like to discuss is the reproduction of the correction to the RT formula [43] when the EWCS extends to the asymptotic boundary. In [9, 60] it was proposed that the BPE (or the reflected entropy) is a good regulator for the entanglement entropy with a geometric cutoff [10]. Let us shrink $A'_1$ and $A'_2$ while keeping $A'_1AA'_2$ fixed, hence the geodesic extended from the EWCS is also fixed. Accordingly, the regions $B'_1$ and $B'_2$ will also shrink due to the balance conditions. When $A$ approaches $AA'$, the EWCS approaches the RT surface such that

$$BPE(A,B) = s_{AA'}(A)|_{balanced} \rightarrow S_A. \tag{125}$$

The physical meaning of the PEE tells us that, the regulator means to ignore the contribution from $A'$ to $S_A$.

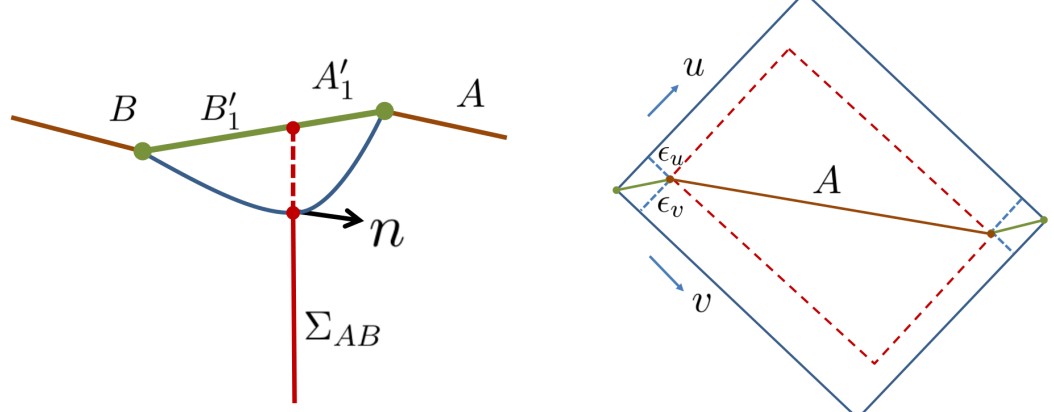

Figure 9: Left: The blue line is the RT surface $\mathcal{E}_{AB}$, while the red curve is the EWCS $\Sigma_{AB}$ which is normal to $\mathcal{E}_{AB}$. The cutoff region $A'_1B'_1$ is now boosted. Right: A causal wedge in the Rindler space with cutoffs $\epsilon_{u,v}$. The green segments are $A'_1$ and $A'_2$.

[10]In [60], the geometric regulator is classified and its difference between the UV cutoff was clarified. Nevertheless, in two-dimensional theories the difference only affects higher order corrections which disappears when $\delta \rightarrow 0$.

Let us consider the configuration shown in Fig.8 and assume the figure is settled on a time slice. According to our standard to choose $v$ and $\tilde{n}$, when the cutoff region $A'_i$ is infinitesimal, we get the same $n$ as [43] near the asymptotical boundary, i.e. $n \propto \partial_t$. Hence in the static case our prescription reproduces the prescription proposed in [43] to calculate the correction to the entanglement entropy.

However, in covariant scenarios, we have the freedom to rotate the cutoff region $A'_i B'_i$ while keeping the causal development of $AA'$ fixed. Such rotations do not change the EWCS but will change the normal vectors $n$ near the asymptotical boundary (see the left figure in Fig.9). The change of the $n$ vector will affect the integral (66) significantly, which indicates that the correction from the CS term is highly *sensitive* to the boost angle of cutoff region $A'_i B'_i$. This goes beyond the scope of the paper [43].

Fortunately, we find that another calculation for such scenarios was carried out in [83], which calculated the correction using the generalized Rindler method [19, 20]. Consider an interval with endpoints being $(-\frac{l_u}{2}, -\frac{l_v}{2})$ and $(\frac{l_u}{2}, \frac{l_v}{2})$, the causal wedge covers

$$-\frac{l_u}{2} < u < \frac{l_u}{2}, \qquad -\frac{l_v}{2} < v < \frac{l_v}{2}, \tag{126}$$

which is captured by the rectangular region with blue boundary in the right figure of Fig.9. One can construct the Rindler transformations which map the causal wedge to an infinitely large Rindler spacetime. Introducing a volume cutoff such that the Rindler space only covers a subset (see the region enclosed by the dashed red rectangular) of the causal wedge

$$-\frac{l_u}{2} + \epsilon_u < u < \frac{l_u}{2} - \epsilon_u, \qquad -\frac{l_v}{2} + \epsilon_v < v < \frac{l_v}{2} - \epsilon_v, \tag{127}$$

where $\epsilon_{u,v}$ are infinitesimal positive parameters. Here the cutoff scheme is the same for the two endpoints. The short green segments in the right figure is just the $A'_1$ and $A'_2$ regions which are now determined by $\epsilon_{u,v}$. The regions $B'_{1,2}$ are determined by $A'_{1,2}$ via the balance conditions. As was classified in [60], the entanglement entropy evaluated by the Rindler method belongs to those evaluated from entanglement contour via geometric regulators. Hence the results from the Rindler method should be consistent with those evaluated using the BPE, the reflected entropy or the EWCS with the CS term correction. The correction to the holographic entanglement entropy from the CS term is evaluated by calculating the area of the inner horizon in the Rindler space [48]. With the regulation parameters $\epsilon_u$ and $\epsilon_v$ given, the result is quite simple [83],

$$S_A^{CS} = \frac{1}{4G\mu} \log\left[ \frac{l_u}{\epsilon_u} \frac{\epsilon_v}{l_v} \right]. \tag{128}$$

In [83] the author made the choice $\epsilon_u = \epsilon_v$ hence the cutoff region $A'_i$ is static. This choice reproduces the result of [43]. Nevertheless, if we keep the cutoff region covariant the result will be quite sensitive to the boost angle of $A'_i$. This sensitivity gives a non-trivial consistency test for our prescription.

In section 4.3, the results for the adjacent cases are based on our choice $n \propto \partial_t$ at the joint endpoint of $A$ and $B$. One can check that these results can only be reproduced from the non-adjacent cases with static $A'_1 B'_1$.

## Acknowledgements

We wish to thank Debarshi Basu, Ling-Yan Hung, Nabil Iqbal, Hongjiang Jiang and Jianfei Xu for helpful discussions and comments on the draft. QW and HZ are supported by the "Zhishan" Scholars Programs of Southeast University.

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
