# Peer review of "Covariant entanglement wedge cross-section, balanced partial entanglement and gravitational anomalies"

_SciPost Physics, doi:SciPost Phys. 13, 056 (2022)_

## Round 1 · Referee Report · Anonymous (Referee 1) · 2022-7-12

Strengths

1 - New interesting results on balanced partial entanglement and its holographic dual;

2 - Pedagogical introduction to the quantities of interest and systematic presentation of the results.

Weaknesses

1 - Connection with other entanglement quantifiers in mixed states to be improved;

2 - The range of validity of some of the results could be clarified and commented more.

Report

In this work, the authors investigate the balanced partial entanglement (PBE), a recently introduced quantity, in two-dimensional conformal field theories (CFTs) and its conjectured holographic dual, the entanglement wedge cross-section (EWCS).
CFTs with and without gravitational anomalies are considered; in the dual bulk description, the presence of gravitational anomalies in the CFT at the boundary is taken into account by considering topological massive gravity (TMG).
Both the BPE and the EWCS are computed in covariant settings.

In all the cases analyzed in the manuscript, both the BPE and EWCS are found to be equal for the same tripartition, supporting their conjectured duality. Moreover, both of them
match exactly the reflected entropy in all the settings considered. This provides a non-trivial test for the purification independence of the BPE.

The authors consider an interesting and, to the best of my knowledge, unexplored generalization of existing results on BPE and EWCS in the context of holographic two-dimensional CFTs.
The presentation is clear and systematic and the various quantities of interest are presented at the beginning of the manuscript in a pedagogical way.
For these reasons, I believe this work meets the criteria for being published in SciPost.
However, in order to further improve the manuscript, I suggest the following minor points be addressed before publication.

Requested changes

  • In the abstract of the work, the authors claim to study the BPE in two-dimensional holographic CFTs, but, from the computation reported in Sec. 3, it is not clear whether their results hold for holographic CFTs only or for generic CFTs. If the latter case is true, I think it is worth stressing it more in the paper and commenting more on this point, given that this result would go beyond the context of AdS/CFT and could in principle be useful also for other communities.

  • Eq. (3.10) relates the BPE to the mutual information. Given that this is an important quantity throughout the section, it would be useful for the reader to spend some words on it and provide some references where its expressions have been obtained.

  • It would be useful to mention which of the previous relations have been used for obtaining (3.12). In this way the reader can follow more easily the logical flow of the section.

  • At the end of Sec. 3 the authors mention a numerical check of the fact that the crossing partial entanglement entropy is minimal at the balance point. Would it be possible for the authors to provide a plot to explicitly show this statement for a given set of parameters?

  • As the authors mentioned in the introduction, a very well-known measure of entanglement in mixed states is the negativity. I believe that, for improving the general framework of the manuscript, it would be greatly insightful providing a discussion on which kind of information on quantum correlations negativity, BPE and reflected entropy contain and, in particular, in which sense such a content is different for these three quantities. Moreover, it would be extremely interesting whether the authors could provide a comparison between the original results for the BPE reported in this manuscript and the existing ones for the negativity in the same scenarios, i.e. in CFTs with and without gravitational anomalies. This would make the work more complete and self-contained.

I have also found some typos in the text, which can be easily fixed. In the following I list some of them.

  • pag.2, 6 lines above (1,1): "Since The negativity" must be replaced by "Since the negativity";

  • pag.2: the sentence before (1,1), which begins with "In the context of holography,...", is not clear and should be rewritten, given that it introduces the notion of entanglement wedge, crucial for the forthcoming discussion;

  • pag.5, 8 lines above (2,1): "corresponce" must be replaced by "correspondence";

  • pag.7: 4 lines before the end of the page, the formula $A'B''=A'\cup B'$ contains probably a typo given that the subsystem $B''$ has never been introduced;

  • pag.8: just above (2,10) you say that $\mathcal{I}_{AB'}$ is half of the mutual information, but in (2.10) $\mathcal{I}_{AB}$ appears;

  • pag.26, two lines below (4.55): "correspond" must be replaced by "corresponds".

  • validity: high
  • significance: good
  • originality: good
  • clarity: high
  • formatting: good
  • grammar: good

Author:  Qiang Wen  on 2022-07-23  [id 2680]

(in reply to Report 1 on 2022-07-12)

We thank the referee very much for the recommendation

1, The referee wrote: In the abstract of the work, the authors claim to study the BPE in two-dimensional holographic CFTs, but, from the computation reported in Sec. 3, it is not clear whether their results hold for holographic CFTs only or for generic CFTs. If the latter case is true, I think it is worth stressing it more in the paper and commenting more on this point, given that this result would go beyond the context of AdS/CFT and could in principle be useful also for other communities.

Our reply: Thanks for pointing this out. We added “(holographic)” in the abstract, as pointed out by the referee, this is misleading. Now the parenthesis is deleted. Also we added the following sentence to the first paragraph of section 3, “Note that, the BPE can be defined in non-holographic systems, hence the calculations in this section goes beyond AdS/CFT.”

2, The referee wrote: Eq. (3.10) relates the BPE to the mutual information. Given that this is an important quantity throughout the section, it would be useful for the reader to spend some words on it and provide some references where its expressions have been obtained.

Our reply: Thanks. We just noticed that the equation (3.6) in the first submission, was not written in a proper way according to the paragraph before (3.10). In the new submission the equation (3.6) is noted as (27) and the BPE is decomposed into three parts. It is then easy to see that, the first term plus the second term in the BPE$(A,B)$ (27) give the half the mutual information $I(A,B)/2=(S_{A}+S_{B}-S_{AB})/2$.

3, The referee wrote: It would be useful to mention which of the previous relations have been used for obtaining (3.12). In this way the reader can follow more easily the logical flow of the section.

Our reply: Paragraphs near (3.12) is written to clarify the logic.

4, The referee wrote: At the end of Sec. 3 the authors mention a numerical check of the fact that the crossing partial entanglement entropy is minimal at the balance point. Would it be possible for the authors to provide a plot to explicitly show this statement for a given set of parameters?

Our reply: Figure 4. is added to illustrate one simple numerical check. We put the 2d figure instead of a 3d figure with the two variables x_3 and t_3, because the saddle in the 3d figure is not very clear to see.

5, The referee wrote: As the authors mentioned in the introduction, a very well-known measure of entanglement in mixed states is the negativity. I believe that, for improving the general framework of the manuscript, it would be greatly insightful providing a discussion on which kind of information on quantum correlations negativity, BPE and reflected entropy contain and, in particular, in which sense such a content is different for these three quantities. Moreover, it would be extremely interesting whether the authors could provide a comparison between the original results for the BPE reported in this manuscript and the existing ones for the negativity in the same scenarios, i.e. in CFTs with and without gravitational anomalies. This would make the work more complete and self-contained.

Our reply: I understand that entanglement negativity now attracts considerable attention from both of the condensed matter community and the high energy community. However it is defined quite differently from the reflected entropy and BPE, hence it is difficult to see how these quantities are related and differ from each other. In AdS/CFT these three quantities are all claimed to be dual to the EWCS. The negativity in CFT$_2$ with gravitational anomaly is also carried out in Ref. [44] (arxiv:2202.00683). However the result does not match with the EWCS. For example in the adjacent case, the negativity differ from the EWCS by a constant of order $c$, which is just the balanced crossing PEE. On the other hand the negativity calculated by the correlation functions of twist operators Ref. [13] (1907.07824) gives a more accurate matching with the EWCS and the reflected entropy. We hope we can return to this point and get further understanding in the future.

We added the above information to the last paragraph in section 5.1

6, We thank the referee for pointing out multiple typos in the paper, they are all fixed in the new submission.

---

## Round 1 · Referee Report · Anonymous (Referee 2) · 2022-7-21

Report

The authors study the covariant balanced partial entanglement (BPE) in 2D CFTs and the covariant entanglement wedge cross section (EWCS) in 3D gravity. Both the cases with and without gravitational anomaly are considered. For BPE, balance conditions separately for the right-moving and the left-moving sectors are imposed. For EWCS, the correction due to the CS term in TMG is evaluated. The results of BPE match the results of EWCS and also match the results of reflected entropy in the literature.

The topic is interesting and the results in this paper give further support to the conjecture that the BPE is purification-independent and is always the same as the reflected entropy. It is a good addition to the study of mixed-state entanglement. I recommend the paper for publication in SciPost Physics.

Requested changes

In the middle of page 4, the so-called ALC proposal is mentioned without any explanation. It would be better to mention on page 4 that the ALC proposal would be explained around eq. (2.7) in the middle of page 7.

  • validity: high
  • significance: high
  • originality: high
  • clarity: high
  • formatting: excellent
  • grammar: excellent

Author:  Qiang Wen  on 2022-07-23  [id 2679]

(in reply to Report 2 on 2022-07-21)

We thank the referee for recommendation, the problem pointed out by the referee has been properly addressed in the new submission.

---

## Round 2 · Referee Report · Anonymous (Referee 1) · 2022-7-24

Report

I thank the authors for having addressed all the points raised in the report. I recommend this manuscript for publication on SciPost.

---

## Round 2 · List of Changes

The main changes are in the following

1, we added a figure at the end of section 3 to give an example to numerically test that the crossing PEE arrives its the minimal value at the balanced point.

2, a paragraph is added at the end of section 5.1 to briefly discuss the relations of the BPE and EWCS with the negativity.

Also there are multiple minor changes according to the request changes from the referee.

---

## Editorial Decision

published